# meQTL mapping in the GENOA study reveals genetic determinants of DNA methylation in African Americans

Lulu Shang [1], Wei Zhao [2], Yi Zhe Wang [2], Zheng Li [1], Jerome J. Choi[3], Minjung Kho[2], Thomas H. Mosley[4], Sharon L. R. Kardia[2], Jennifer A. Smith [2] ✉ & Xiang Zhou [1] ✉

Identifying genetic variants that are associated with variation in DNA methylation, an analysis commonly referred to as methylation quantitative trait locus (meQTL) mapping, is an important first step towards understanding the genetic architecture underlying epigenetic variation. Most existing meQTL mapping studies have focused on individuals of European ancestry and are underrepresented in other populations, with a particular absence of large studies in populations with African ancestry. We fill this critical knowledge gap by performing a large-scale cis-meQTL mapping study in 961 African Americans from the Genetic Epidemiology Network of Arteriopathy (GENOA) study. We identify a total of 4,565,687 cis-acting meQTLs in 320,965 meCpGs. We find that 45% of meCpGs harbor multiple independent meQTLs, suggesting potential polygenic genetic architecture underlying methylation variation. A large percentage of the cis-meQTLs also colocalize with cis-expression QTLs (eQTLs) in the same population. Importantly, the identified cis-meQTLs explain a substantial proportion (median = 24.6%) of methylation variation. In addition, the cis-meQTL associated CpG sites mediate a substantial proportion (median = 24.9%) of SNP effects underlying gene expression. Overall, our results represent an important step toward revealing the co-regulation of methylation and gene expression, facilitating the functional interpretation of epigenetic and gene regulation underlying common diseases in African Americans.

Genome-wide association studies (GWAS) have identified thousands of genetic variants that are associated with various diseases and disease-related complex traits. However, the majority of these disease-associated variants reside in noncoding regions and have unknown functions[1-4]. Although variants in noncoding regions cannot directly influence the function of a gene by disrupting protein-coding sequence[4-6], they can influence gene expression through epigenetic regulatory mechanisms or induce epigenetic changes through the regulation of gene expression[7-9]. One important epigenetic change is DNA methylation, which influences gene expression by altering transcription factor binding capability, inducing conformational changes in the chromatin via histone modifications, and regulating microRNA

[1]Department of Biostatistics, School of Public Health, University of Michigan, Ann Arbor, MI 48109, USA. [2]Department of Epidemiology, School of Public Health, University of Michigan, Ann Arbor, MI 48109, USA. [3]Population Health Sciences, University of Wisconsin-Madison School of Medicine and Public Health, Madison, WI 53726, USA. [4]Memory Impairment and Neurodegenerative Dementia (MIND) Center, University of Mississippi Medical Center, Jackson, MS 39126, USA. ✉e-mail: smjenn@umich.edu; xzhousph@umich.edu

expression[10,11]. Therefore, identifying genetic variants that are associated with variation in DNA methylation through methylation quantitative trait locus (meQTL) mapping becomes an important first step towards understanding the epigenetic mechanisms underlying disease associations and towards investigating the genetic architecture underlying interindividual differences in the epigenome. In recent years, meQTL mapping studies have successfully identified many cis-acting genetic variants that are associated with DNA methylation levels[9,12–14]. Many of these identified meQTLs are colocalized with other molecular QTLs such as eQTLs that influence gene expression, supporting the regulatory functions of meQTLs[12,15–18]. Importantly, some meQTLs are also associated with behavioral and disease-related complex traits such as smoking, aging, body mass index, and type 2 diabetes[19], highlighting the importance of meQTL for understanding the genetic and epigenetic mechanisms underlying diseases[20].

Most existing meQTL mapping studies have thus far focused on individuals of European ancestry and are noticeably underrepresented in other populations, with a particular absence of large studies in populations with African ancestry. However, the meQTLs observed in one population are not necessarily meQTLs in another population due to differences in allele frequencies and linkage disequilibrium patterns as well as differences in the genetic architecture underlying methylation variation across ancestries. In addition, of the few existing meQTL mapping studies performed in African ancestry participants, most had small sample sizes and reported weak associations. For example, one meQTL mapping study examined lymphoblastoid cell lines from 77 HapMap Yoruba samples[18]. Another study collected 90 peripheral blood samples and 87 umbilical cord blood samples from African American participants[21]. A handful of studies have also evaluated whether meQTLs identified in European ancestry populations are replicated in African Americans. For example, results of a replication meQTL mapping study in the Framingham Heart Study (FHS)[20] were investigated in 384 African Americans from the Grady Trauma Project (GTP)[22,23]. Sample sizes of African ancestry populations in these studies remain relatively small as compared to other populations. For example, the Genetics of DNA Methylation Consortium (GoDMC) included 27,750 individuals from European ancestry[24], and Hawe et al. included 3799 Europeans and 3195 South Asians[25] for meQTL mapping. Lack of large-scale, well-powered meQTL mapping studies in populations with African ancestry can impede our understanding of the epigenetic mechanisms underlying DNA methylation, gene expression, and disease etiology for these populations.

In this work, to fill this critical gap, we pair genotyped and methylation data from 961 African Americans from the Genetic Epidemiology Network of Arteriopathy (GENOA) study to perform a comprehensive cis-meQTL mapping analysis. In addition, we integrate the meQTL mapping results with eQTL mapping results in the same sample[26] through colocalization and mediation analysis. Overall, our results represent an important step towards revealing the shared regulatory roles of methylation and gene expression, facilitating the functional interrogation of epigenetic and gene regulatory mechanisms underlying diseases in African Americans.

## Results
### meQTL mapping in African Americans
We focused on methylation measurements on 771,134 CpG sites from 961 African American (AA) samples in the GENOA study (Supplementary Fig. 1). The methylation level across CpG sites displays an expected bimodal distribution pattern. Specifically, 28.57% of CpG sites have a beta value below 0.3 and are centered around a mode near 0.109, representing hypomethylated CpG sites that are unmethylated in the majority of samples. And 61.96% of CpG sites have a beta value above 0.7 and are centered around a mode near 0.96, representing hypermethylated CpG sites that are methylated in the majority of samples[27]. The hypomethylated sites are mainly located within transcription start

sites and/or CpG islands (TSS1500: OR = 2.51, $p$-value < 2.23e−308; TSS200: OR = 12.14, $p$-value < 2.23e−308; CpG islands: OR = 21.59, $p$-value < 2.23e−308), while the hypermethylated sites are located in other regions of the genome (intergenic region: OR = 1.78, $p$-value < 2.23e−308; gene body: OR = 3.04, $p$-value < 2.23e−308; CpG open sea: OR = 6.54, $p$-value < 2.23e−308) (Supplementary Fig. 2). The results suggest that the hypomethylation near the promoter and the transcription start site may reflect accessible chromatin, binding of TFs, and active transcription[28].

In total, we identified 320,965 meCpGs and 4,565,687 cis-meQTLs (Supplementary Table 1), with a few most significant examples shown in Supplementary Fig. 3. The results are largely similar when we used different window sizes of cis-SNPs for cis-meQTL mapping analysis (Supplementary Figs. 4–8 and Supplementary Table 2). The results are also largely similar when we varied the number of permutations used to construct the empirical significance cutoff (Supplementary Fig. 9). The identified meCpG sites are enriched with hypermethylated CpG sites (OR = 1.05, $p$-value < 2.23e−308) and are depleted with hypomethylated CpG sites (OR = 0.56, $p$-value < 2.23e−308), where hypermethylated and hypomethylated CpG sites are defined as CpG sites with a beta value above 0.7 or below 0.3, respectively[29]. The identified meQTLs are strongly enriched near the associated CpG sites (Fig. 1a) and can increase methylation at some CpGs while decreasing methylation at others. Specifically, among the 4,565,687 unique meQTLs, the number of CpGs associated with an meQTL ranges from 1 to 166 (median = 2, mean = 3.4). The number of CpGs positively associated with an meQTL ranges from 0 to 118 (median = 1, mean = 1.7) while the number of CpGs negatively associated with an meQTL ranges from 0 to 83 (median = 1, mean = 1.7). The median proportion of positively associated CpGs for an meQTL is 0.5 (mean = 0.49, Supplementary Fig. 10).

The identified meQTLs in the present study replicate a large proportion of the meQTLs identified in previous studies of different genetic ancestries, regardless of significance thresholds used (Supplementary Table 3). Specifically, we calculated the replication rate ($\pi_1$), which effectively represents the proportion of signals detected in the previous study that are replicated in the present study. The replication rate in GENOA ranges from 0.9 to 0.93 at different cutoffs when compared to the Hawe et al. study with European and South Asian ancestries[25]. The meQTLs detected in either European or South Asian ancestry of the Hawe et al. study but not replicated in GENOA tend to have lower absolute effect sizes (median = 0.208 in comparison with EU and 0.198 with SA) than those detected in both studies (median = 0.558 in comparison with EU and 0.538 with SA, Supplementary Fig. 11). The meQTLs detected in European or South Asian but not in GENOA also tend to have a lower allele frequency (median = 0.166 in comparison with EU, 0.17 with SA) to those detected in both studies (median = 0.244 in comparison with EU, 0.247 with SA, Supplementary Fig. 11). In addition, the replication rate of GENOA ranges from 0.9 to 0.93 at different significant $p$-value cutoffs when compared to the BEST study of South Asian ancestry[15]. The meQTLs detected in BEST but not in GENOA tend to have lower absolute effect sizes (median = 0.214) in GENOA than those detected in both studies (median = 0.633, Supplementary Fig. 12). The meQTLs detected in South Asian individuals in BEST but not in GENOA tend to have a lower allele frequency (median = 0.223) to those detected in both studies (median = 0.274, Supplementary Fig. 12). The replication rate of GENOA ranges from 0.76 to 0.77 at different significant $p$-value cutoffs when compared to the GoDMC study of European ancestry[24] (Supplementary Table 3). The meQTLs detected in GoDMC but not in GENOA tend to have lower absolute effect sizes (median = 0.149) in GENOA than those detected in both studies (median = 0.541, Supplementary Fig. 13). The allele frequencies in European individuals for the SNPs in SNP-CpG pairs detected in GoDMC but not in GENOA tend to have a lower allele frequency (median = 0.171) to those detected in both studies

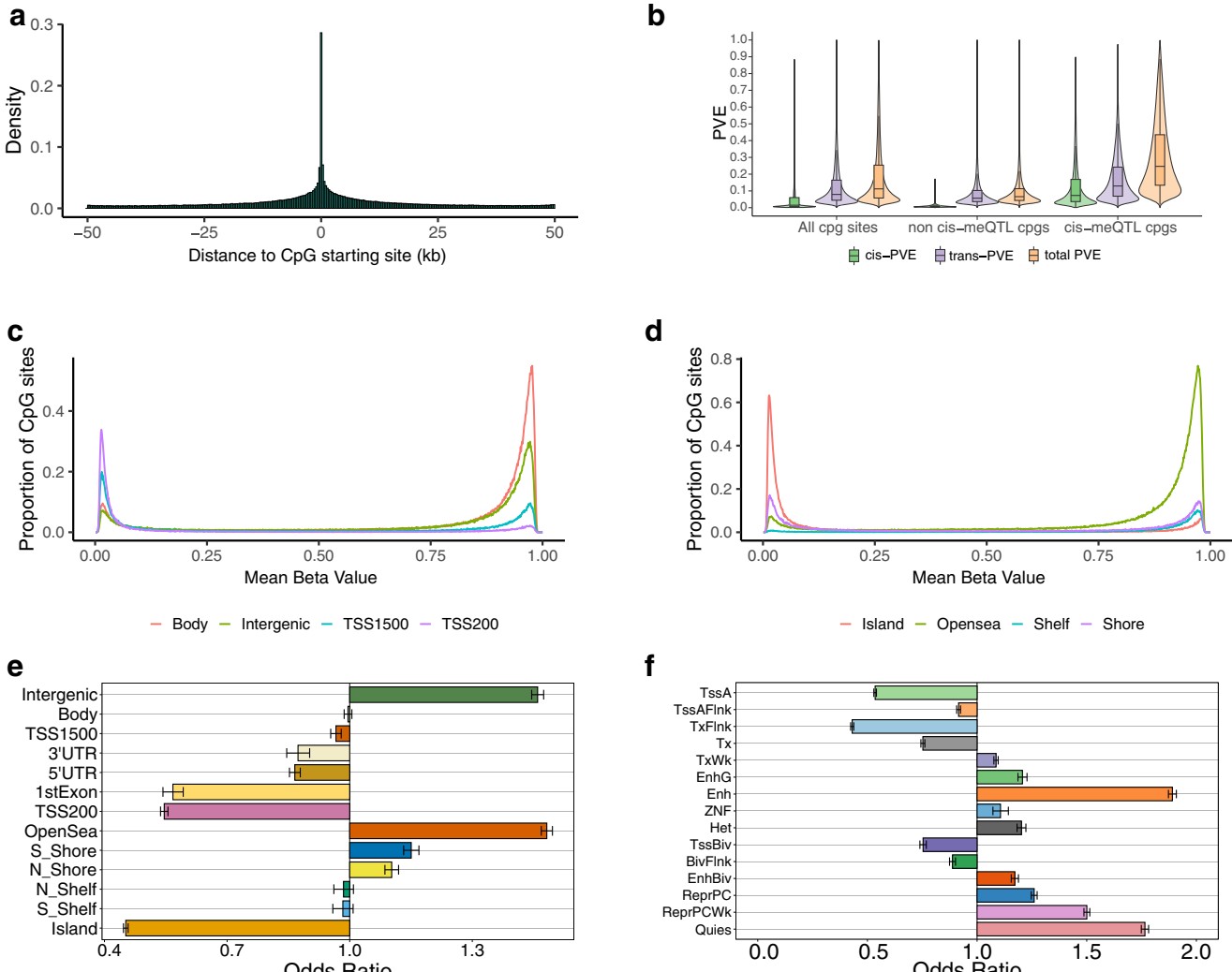

**Fig. 1 | Characterization of the identified meQTLs. a** The locations of the meQTLs are shown relative to the CpG sites. **b** Proportion of variance (PVE; aka SNP heritability) in methylation levels estimated for all CpG sites (left three plots), CpGs without detectable meQTLs (middle three plots), and CpGs with detectable meQTLs (right three plots). PVE explained by all SNPs is referred to as total PVE (orange), which is partitioned into a component that is explained by cis-SNPs (cis-PVE; green) and another component that is explained by trans-SNPs (trans-PVE; purple). In the boxplots, the center line, box limits and whiskers denote the median, upper and lower quartiles, and 1.5× interquartile range, respectively. **c** Mean beta value distribution stratified by locations relative to gene regions. Beta values are the ratio of the normalized intensity of the methylated bead type to the combined normalized locus intensity, which range from 0 (hypomethylated) to 1 (hypermethylated). The gene body and intergenic regions are more hypermethylated, the

TSS1500 and TSS200 are more hypomethylated. **d** Mean beta value distribution stratified by locations relative to CpG islands. The open sea is more hypermethylated and the CpG island is more hypomethylated. **e** Enrichment odds ratio of meCpGs, together with the 95% CI, is computed for CpG sites residing in different annotated genomic regions. CpG sites with meQTLs are enriched in intergenic region, gene body and open sea regions, and depleted in TSS regions, 3' and 5' UTR, 1st exon and CpG island. Bar graphs show odds ratios; error bars show 95% confidence intervals. **f** Enrichment odds ratio of meCpGs, together with the 95% CI, is computed for CpG sites residing in 15 different chromatin states as defined by the Roadmap Epigenomics project through ChromHMM in blood samples. Bar graphs show odds ratios; error bars show 95% confidence intervals. Statistics were computed based on a sample size of $n = 961$ and for 728,578 CpG sites.

(median = 0.245, Supplementary Fig. 13). Certainly, a lack of complete replication among different studies is expected, given that statistical power is unlikely to be fully achieved in any study and that different studies differ in terms of genetic ancestry, methylation measurement platform (27K[21], 450K[15,20], or EPIC BeadChip), cis window size (1 Mb[20], 500 kb[15], or 50 kb[21]), types of tissue used (whole blood[15,20], peripheral blood[21], or cord blood[21]), sample sizes, as well as the applied FDR methods (permutation based[15,20] or Holm correction[21]).

## SNP heritability underlying methylation

We examined the genetic architecture underlying methylation level variation through heritability estimation and partitioning. First, for each CpG site in turn, we estimated the proportion of variance (PVE) in methylation levels that are accounted for by all SNPs, a quantity

commonly referred to as SNP heritability, using the Bayesian sparse linear mixed model (BSLMM). We found that the median PVE estimate across all CpG sites is 11.24% (mean estimate = 18.97%, sd = 18.8%). As one might expect, meCpGs tend to have a higher PVE than non-meCpGs (p-value < 2.23e−308): the median PVE is 24.64% across meCpGs (mean = 30.58%, sd = 21.14%) and is 6.57% across non-meCpGs (mean = 9.72%, sd = 9.19%) (Supplementary Fig. 14). Next, we used BSLMM to partition the PVE of each CpG site into two parts: one that is explained by cis-SNPs (i.e., cis-PVE) and the other that is explained by trans-SNPs (i.e., trans-PVE). Consistent with refs. 30,31, we found that the majority of PVE in methylation level is explained by trans-SNPs, with only a fraction explained by cis-SNPs. Specifically, the median proportion of PVE explained by cis-SNPs is only 1.47% (mean = 6.33%; sd = 11.48%) across all CpG sites, with the remaining explained by trans-

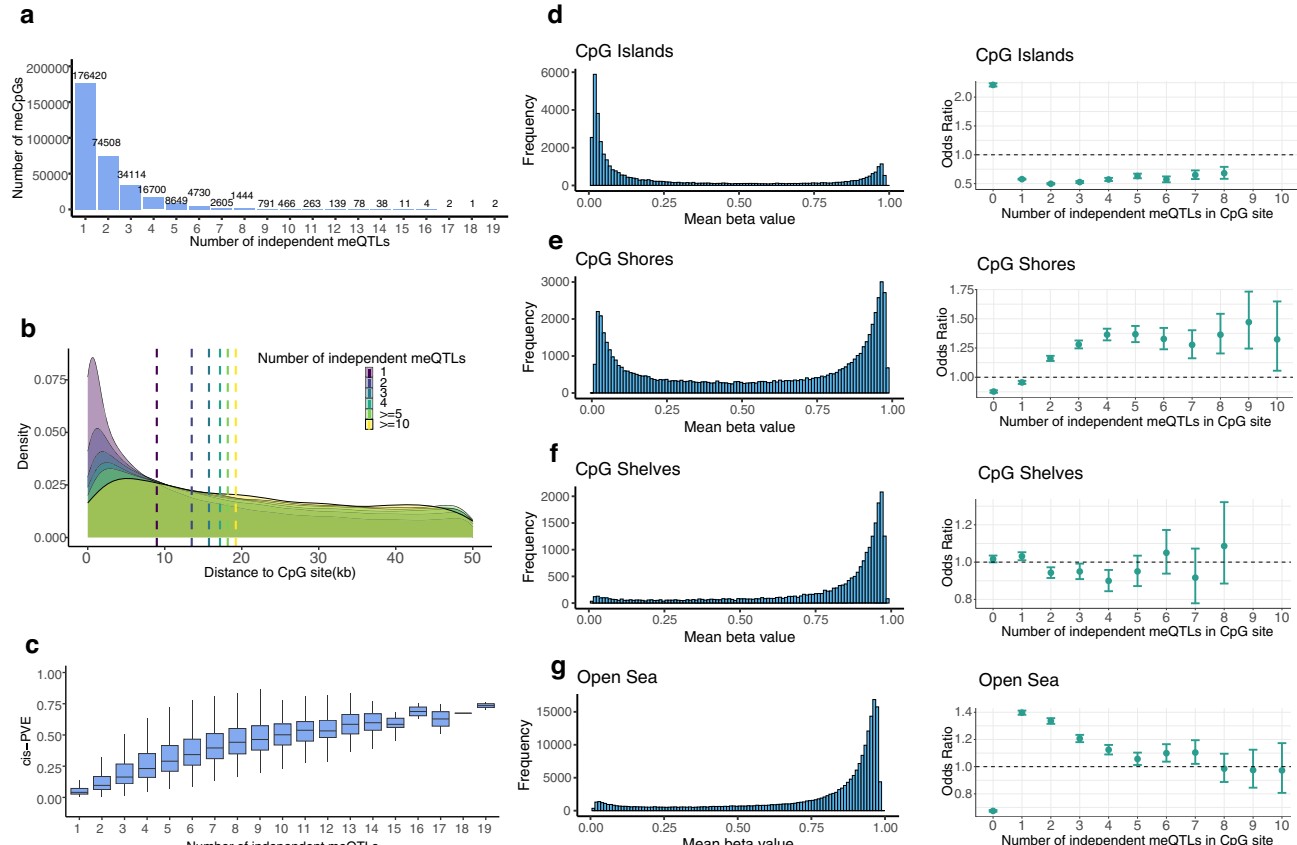

**Fig. 2 | Characterization of the conditional meQTLs. a** Histogram shows the number of meCpGs (y axis) that harbor different numbers of independent meQTLs (x axis). We displayed meCpGs that harbor up to 19 independent meQTLs, with the detailed number of meQTLs listed above each bar. A large fraction of meCpGs harbor a small number of independent meQTLs. **b** Density plot shows the distance from meQTL to the corresponding meCpG. The density plot is stratified by the number of meQTLs: meCpGs with one, two, three, four, five or more, and 10 or more independent meQTLs. Dashed lines represent the median distance between meQTL and meCpG in the six stratified groups. **c** The proportion of variance (PVE) in methylation levels explained by SNPs are higher for meCpGs that harbor a larger number of independent meQTLs. In the boxplots, the center line, box limits and whiskers denote the median, upper and lower quartiles, and 1.5× interquartile range, respectively. **d**–**g** Left: Histograms of mean beta value distribution stratified by locations relative to CpG islands (**d**: CpG islands; **e**: CpG shores; **f**: CpG shelves; **g**: Open sea). Right: Enrichment odds ratios of CpG sites in different functional groups. The CpG sites are stratified by the number of meQTLs. In each stratified set of CpGs in the functional groups, we only calculate odd ratios for the sets with at least 100 CpGs. Error bars show 95% confidence intervals. Statistics were computed based on a sample size of $n = 961$ and for 728,578 CpG sites.

SNPs (median = 7.91%; mean = 12.64%; sd = 12.14%). As one might expect, cis-SNPs explain a higher proportion of cis-PVE in meCpGs (median = 7.29%) than in non-meCpGs (median = 0.697%) (Fig. 1b).

**Conditional analysis revealed additional cis-meQTLs**
Because the primary meQTL only explains a median of 68.62% of the cis-PVE in the meCpGs, we performed conditional analysis with forward-backward stepwise regression to identify additional meQTLs that are independent of the primary ones. In total, we identified a total of 609,992 independent meQTLs through conditional analysis (Fig. 2a and Supplementary Table 4), which include 289,027 conditional meQTLs on top of the primary meQTLs previously identified. We found that most meCpGs (54.97%) contain only one independent meQTL (i.e., primary meQTL). A substantial proportion (23.21%) of meCpGs contain two independent meQTLs, and the remaining meCpGs contain three or more independent meQTLs (21.82%). The conditional meQTLs reside further away from the CpG site compared to the primary meQTLs, though they are still enriched around the CpG site when compared with non-meQTLs (i.e., SNPs that are not significant for any SNP-CpG pairs tested) (Fig. 2b). The number of meQTLs in the standard and conditional analyses are positively correlated with each other (Pearson's correlation coefficient = 0.347, *p*-value < 2.23e −308). The number of independent meQTLs across meCpGs is also

positively correlated with the cis-PVE of each meCpGs (Pearson's correlation coefficient = 0.593, *p*-value < 2.23e−308; Fig. 2c). By using conditional meQTLs in addition to primary meQTLs, a higher proportion of cis-PVE (87.35%) is explained compared to that explained by using primary meQTLs alone (Supplementary Figs. 15–17).

**Functional characterization of meCpGs and meQTLs**
We performed enrichment analysis to examine the functional characteristics of the meCpG sites and their enrichment in specific functional genomic regions. In the analysis, we found that meCpG sites are significantly enriched in the intergenic regions (OR = 1.46, *p*-value < 2.23e−308). They are also significantly depleted in TSS1500 (OR = 0.97, p-value = 3.97e−7), TSS200 (OR = 0.55, *p*-value < 2.23e−308), 5'UTR (OR = 0.87, *p*-value = 7.58e−77), 1st exon (OR = 0.57, *p*-value = 2.34e−154) and 3'UTR (OR = 0.87, *p*-value = 8.41e−17) (Fig. 1e). In addition, the meCpG sites are enriched in the open sea (OR = 1.48, *p*-value < 2.23e−308), the north and south shores (OR = 1.14, *p*-value = 51.8e−106), different sets of enhancer regions characterized by chromHMM (EnhG: OR = 1.21, *p*-value = 5.76e−103; Enh: OR = 1.89, *p*-value < 2.23e−308; EnhBiv: OR = 1.17, *p*-value = 8.0e−111), as well as in weakly transcribed regions (TxWk: OR = 1.09, *p*-value = 2.62e−71). The meCpG sites are depleted in the CpG island (OR = 0.45, *p*-value<2.23e −308) (Fig. 1e), as well as in different sets of promoter regions (TssA:

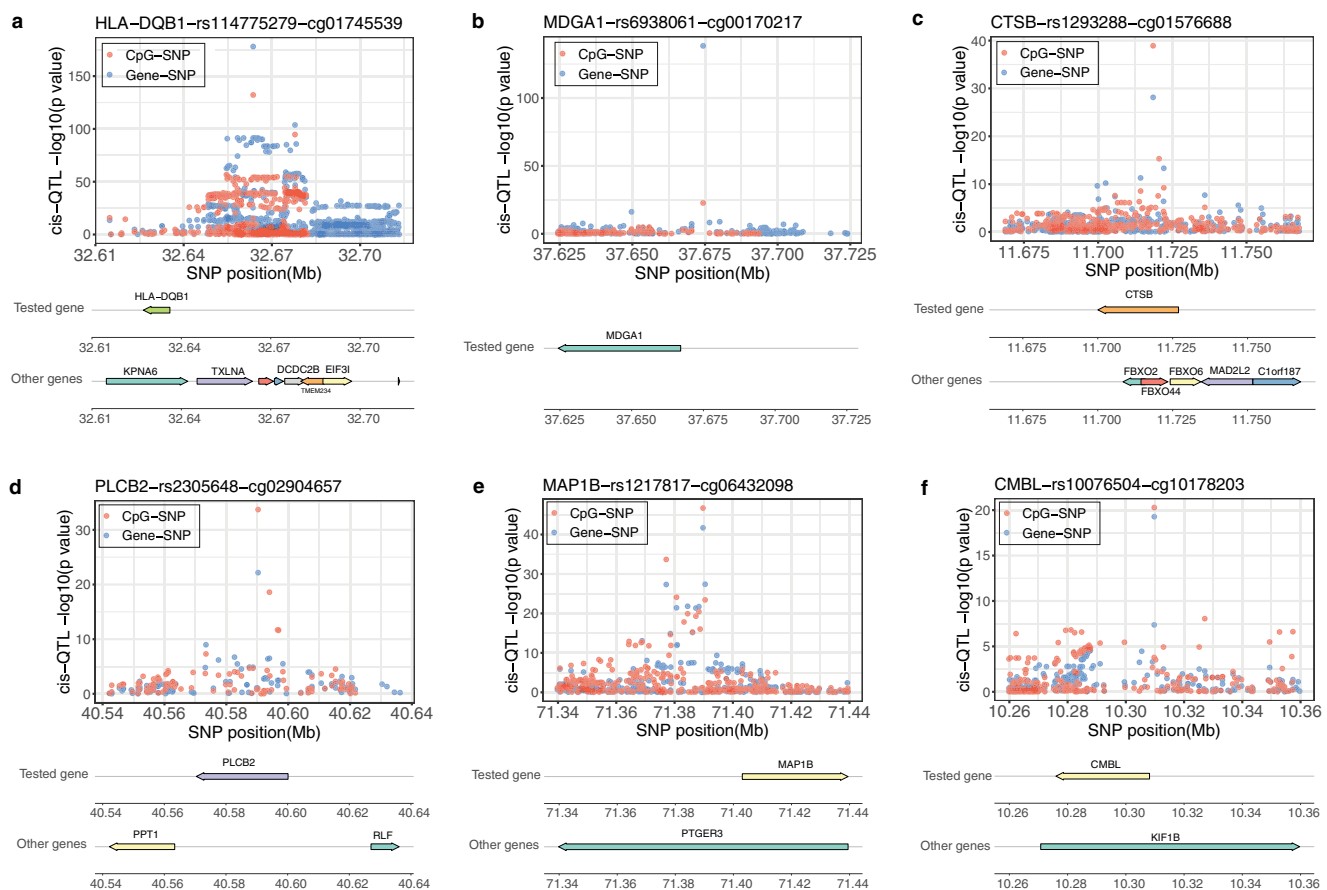

**Fig. 3 | Examples of top six colocalized eGene-meCpG pairs. a–f** *P*-values for the colocalizing eQTL (blue) and meQTL (red) are plotted against physical position. The genes in the tested eGene and meCpG pairs are plotted under the *p*-value figures. The first row is the test gene location, and the second row are other genes inside the given range. Associations were tested by two-sided Wald tests based on a sample size of $n = 961$ for cis-meQTLs and $n = 1032$ for cis-eQTLs.

OR = 0.54, *p*-value < 2.23e−308; TssAFlnk: OR = 0.92, *p*-value = 1.23e−70; TxFlnk: OR = 0.43, *p*-value < 2.23e−308; TssBiv: OR = 0.75, *p*-value = 5.08e−182) and strongly transcribed regions (Tx: OR = 0.75, *p*-value < 2.23e−308, Fig. 1f). The over-representation and under-representation results of the identified cis-meCpG sites in distinct genomic regions in African Americans in the present studies are consistent with the results on individuals from African ancestry[32] as well as European ancestry[24,33–36] in prior studies. We performed meQTL enrichment analysis with either Fisher's exact test or Torus[37] and found the meQTLs are enriched for active chromatin states, such as active transcription start site (TSS)-proximal promoter states (TssA, TssAFlnk), and enhancer states (Enh, EnhG) (Supplementary Fig. 18).

The meCpGs with different numbers of independent meQTLs from the conditional analysis also exhibit distinct enrichment patterns in the shores of CpG islands (Supplementary Fig. 19). Specifically, the meCpGs with only one independent meQTL are depleted in the shores, but the meCpGs with two or more independent meQTLs are enriched in the shores. The number of independent meQTLs and the odds ratio enrichment of the corresponding meCpGs in the shores are positively correlated (Pearson's correlation coefficient = 0.724, *p*-value = 0.018, Fig. 2e). In addition, we found the number of independent meQTLs and the odds ratio of meCpGs are positively correlated in CpG islands (Pearson's correlation coefficient = 0.79, *p*-value = 0.02, Fig. 2d), negatively correlated in open sea (Pearson's correlation coefficient = −0.929, *p*-value = 0.0001, Fig. 2g).

We carefully explored a molecular mechanism through which genetic variants may influence methylation. If a SNP disrupts a transcription factor binding site, then it has a potential to influence methylation of a neighboring CpG site either directly or indirectly

through signaling cascades[9,28,32,38]. Among the 4,565,687 unique meQTLs, 43,115 (0.944%) disrupt TF binding motifs. Among the 609,992 independent meQTLs obtained from conditional analysis, 7445 (1.22%) disrupt TF binding motifs. Consequently, we would expect meQTL enrichment in the set of SNPs that directly disrupt motif binding, as is indeed observed in the present study (OR = 1.16, *p*-value = 2.52e−91).

## Co-localization of eQTLs, meQTLs, and GWAS variants

We explored whether a common/shared genetic variant may influence both gene expression levels and methylation levels. To so do, we first obtained a list of 5475 primary eQTLs previously identified in the GENOA AA gene expression study[26]. We identified the associated genes and CpG sites with each of these eQTLs to form 4854 eGene-meCpG pairs. For each eGene-meCpG pair in turn, we estimated the probability that a common SNP is associated with both gene expression and methylation through co-localization analysis with coloc[39]. We found that a substantial proportion of the tested eGene-meCpG pairs (46.3%) share a common SNP that influences both gene expression and methylation (Supplementary Table 5), with the top examples shown in Fig. 3. The SNP effects on methylation and expression are often in the opposite directions (53%) in the tested eGene-meCpG pairs, and slightly more so (55.4%) in the colocalized pairs (Fig. 4). The CpG sites in the colocalized pairs where the common SNP displays opposite effects on expression and methylation are enriched in promoter regions (OR = 1.46, *p*-value = 9.17e−7), more so than those with the same direction of effects (OR = 1.08, *p*-value = 0.35), as compared to the CpG sites in all tested eGene-meCpG pairs used for colocalization analysis.

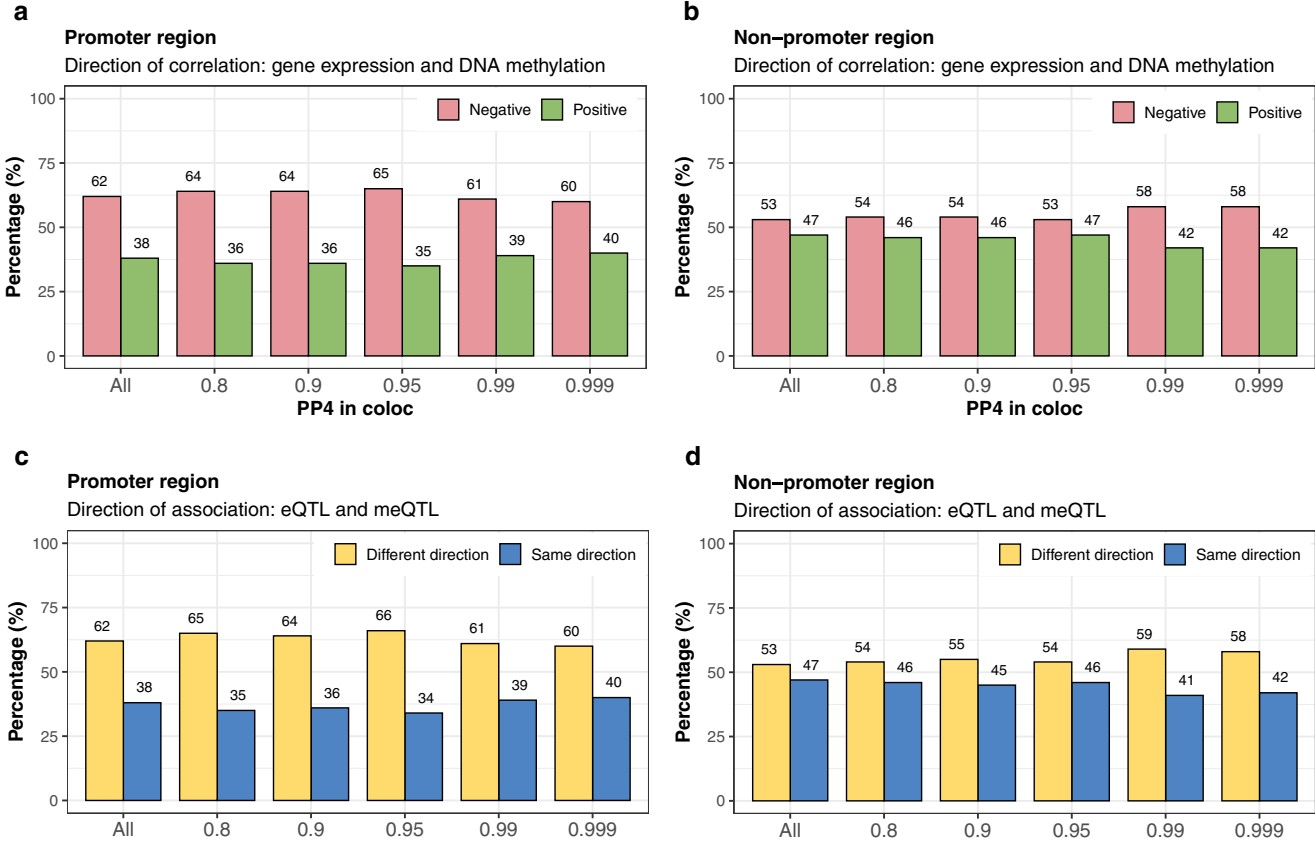

**Fig. 4 | Direction of QTL effects and associations between expression and methylation for co-localized eGene-meCpG pairs.** Results for 4854 eGene-meCpG pairs tested for colocalization using a $p_{12}$ value of 6.3e−04 are presented. Results are stratified according to promoter region and non-promoter region, as well as posterior probability for a shared eSNP in the pairs greater than 0.8, 0.9, 0.95, 0.99, and 0.999. **a**, **b** Histograms of the percentage of eGene-meCpG pairs for which the direction of association between gene expression and DNA methylation is positive or negative in promoter region (**a**) or non-promoter region (**b**). **c**, **d** Histograms of the percentage of eGene-meCpG pairs showing the same or different direction of association in promoter region (**c**) or non-promoter region (**d**).

We obtained summary statistics from GWAS conducted in African Americans for six traits (SBP, DBP, PP, HTN, T2D, and BMI)[40–42] and performed multi-trait colocalization analysis on the tested eGene-meCpG pairs. Multi-trait colocalization examines one GWAS trait at a time and investigates whether the same SNP is associated with gene expression, methylation, and the GWAS trait. A positive multi-trait colocalization signal would suggest a potentially causal SNP association underlying all three phenotypes as opposed to genetic confounding such as LD between the molecular QTLs and the GWAS association. In the analysis, we identified four multi-trait colocalization signals for four GWAS traits (1 for SBP, 1 for PP, and 2 for DBP, Supplementary Tables 6–7), a number similar to previous studies of other traits[43–46]. As an example, we identified a common SNP, rs2272007, that is associated with the methylation level of cg05589743 (coefficient = −0.375, $p$-value = 5.97e−15), the expression level of *ULK4* (coefficient = −0.994, $p$-value = 1.48e−115), and DBP (coefficient = 0.328, $p$-value = 4e−12, obtained from ref. [47]). rs2272007 is known to be associated with *ULK4* in lymphoblastoid cell lines and with DBP[48]. The CpG site cg05589743 is also located in the 5' UTR region of *ULK4*, which is associated with DBP in African Americans[49]. *ULK4* is a well-known autophagy associated gene[50]. Excessive autophagy can eliminate cellular elements and may cause cell death, and contribute to hypertension-related heart disease[51]. As another example, we found that a common SNP, rs6717671, is associated with the methylation level of the CpG site cg22495460 (coefficient = −0.329, $p$-value = 5.22e−8) located in the gene body, the expression level of *ADCY3* (coefficient = 0.339, $p$-value = 7.65e−9), and the risk of BMI (coefficient = −0.036, $p$-value = 5.35e−5, obtained from ref. [41]). Genetic variation in *ADCY3* is associated with BMI in African Americans[41]. The *ADCY3* gene encodes an enzyme that converts the ATP to cAMP, is involved in a large number of physiological metabolic processes[52], and is an obesity-risk gene reported by previous GWAS in the African Americans[53].

We identified four meQTL-GWAS colocalization signals using coloc (Supplementary Tables 6–7). One of them is also identified in the eQTL-GWAS colocalization analysis and another one is also identified in the GWAS-eQTL-meQTL colocalization analysis. We also identified six meQTL-GWAS colocalization signals using Susie. Five of them are also identified in the eQTL-GWAS colocalization analysis while none of them is supported by multi-colocalization between GWAS-eQTL-meQTL. We further expanded our analysis to using genome-wide meQTLs and identified 0, 8, 1, 4, 1, and 0 colocalized GWAS-meQTL signals for SBP, BMI, DBP, HTN, PP, and T2D, respectively. None of these co-localized meQTLs are associated with gene expression levels. In the ancestry matching colocalization analysis, we identified one multi-trait colocalization with the European ancestry GWAS for BMI. We also identified five GWAS-meQTL colocalization signals with the European Ancestry GWAS (2 for BMI and 3 for DBP). However, the number of identified signals was quite small, suggesting that much larger GWAS of African ancestry are needed in the future to arrive at any definitive conclusions (Supplementary Table 8).

## Mediation analysis
We performed mediation analysis on the 2,246 colocalized eGene-meCpG pairs to further examine the extent to which the shared genetic

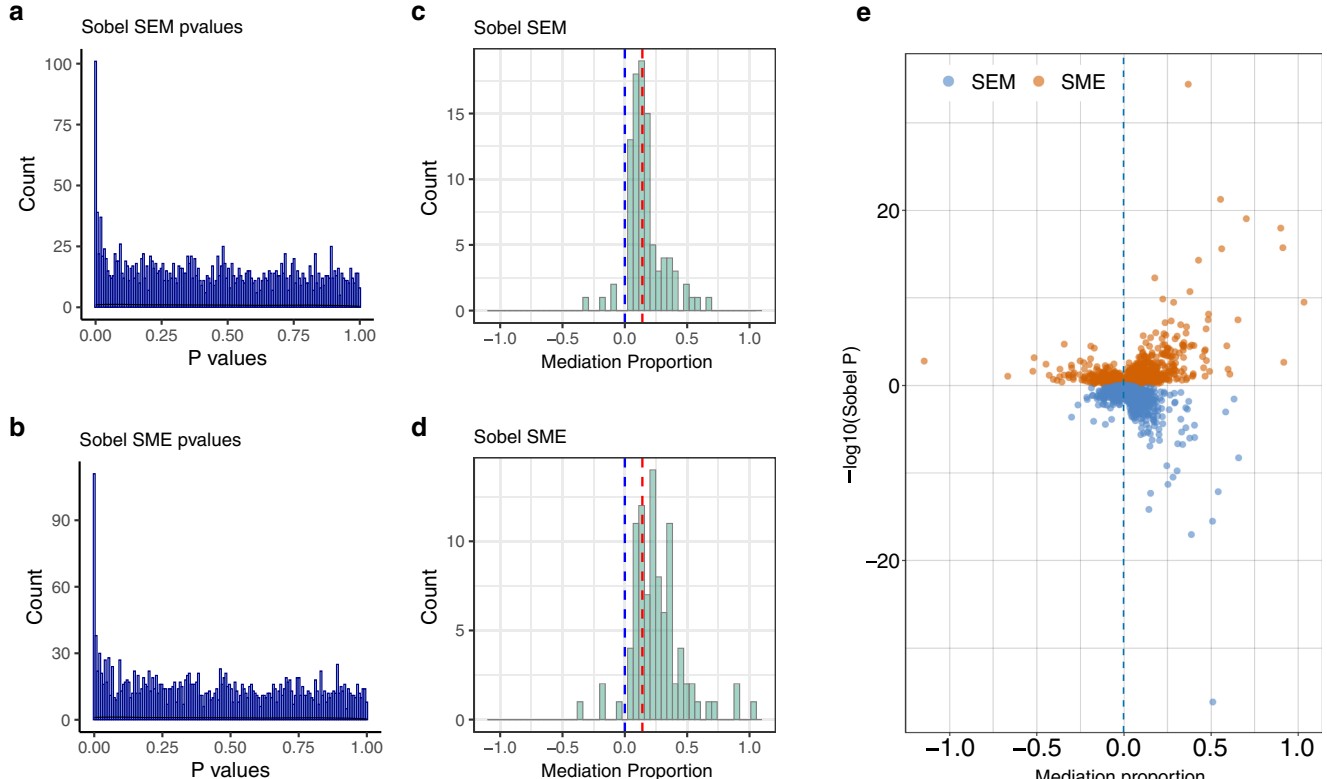

**Fig. 5 | Mediation analyses provide evidence for shared regulatory mechanisms.** Results for two-sided Sobel test on 2246 potentially co-localized eGene-meCpG pairs identified using a $p_{12}$ value 6.3e−04 are presented. **a**, **b** Histograms of two-sided Sobel test p-values for SEM (**a**) and SME (**b**). **c**, **d** Histograms of mediation proportion in significant SEM (**c**) and SME (**d**) (FDR < 0.05) two-sided Sobel tests. Blue vertical line: mediation proportion is 0; red vertical line: median mediation proportion. **e** Mediation analysis results for both the SME (red) and SEM (blue) two-sided Sobel tests. SME: SNP-Methylation-Expression direction; SEM: SNP-Expression-Methylation direction.

variant influences one molecular trait via the other. To do so, we performed two types of mediation analyses to assess evidence that (1) DNA methylation mediates the effect of the genetic variant on gene expression (SNP-Methylation-Expression analysis or "SME") and/or (2) gene expression mediates the effect of the genetic variant on DNA methylation (SNP-Expression-Methylation analysis or "SEM"). At an FDR of 0.05, we identified a total of 111 pairs with significant evidence of mediation in at least one direction (Fig. 5). Among the detected pairs, 103 of them are significant in the SME analysis, 93 are significant in the SEM analysis (Fig. 5), and 85 are significant in both analyses (Supplementary Fig. 20). Among the 103 significant pairs detected in the SME analysis, 95 have a positive mediation proportion, with a median mediation proportion estimate of 0.249 (mean = 0.296) (Fig. 5d). Among the 93 significant pairs of SEM analysis, 89 have positive mediation proportion, with a median mediation proportion estimate of 0.143 (mean = 0.181) (Fig. 5c).

We identified two eGene-meCpG pairs that have an adjusted $q$ value < 0.05 in the SME analysis but have an adjusted $q$ value > 0.1 in the SEM analysis: *TUBD1*/cg02095219 and *CHKB-CPT1B*/cg00047287. In particular, the CpG cg02095219 is located in the open sea near the *TUBD1* gene, and is significant in mediating the effect of rs111282327 on *TUBD1* (SME $p$-value = 0.002, adjusted $q$ value 0.04, Fig. 6a), whose expression is associated with copy number status in primary breast tumors[54]. The CpG site cg00047287 is located in the CpG island and TSS200 of the *CPT1B* gene, and is significant in mediating the effect of rs8137478 on *CHKB-CPT1B* (SME $p$-value = 0.0015, adjusted $q$ value 0.03, Fig. 6b), which is a potential target for age-dependent intramuscular lipid accumulation and insulin resistance[55]. In addition, we also identified one eGene-meCpG pair that has an adjusted $q$ value > 0.1 in the SME analysis but has an adjusted $q$ value < 0.05

in the SEM analysis: *MB21D2*/cg08100000. In particular, the CpG site cg08100000 is located in the gene body of *MB21D2* and is significant in mediating the effect of rs13069487 on cg08100000 (SEM $p$-value = 0.0093, adjusted $p$-value 0.024, Fig. 7). The *MB21D2* gene was identified to be a differentially methylated region (DMR) for human squamous cell carcinoma[56], and overexpression of *MB21D2* can facilitate cell proliferation and invasion in cancer[57].

## Discussion

We have presented a comprehensive meQTL mapping study on 728,578 CpG sites and 8,993,056 unique cis-SNPs in 961 African Americans. We have identified a total of 4,565,687 cis-meQTLs and 320,965 meCpG sites[20,58], as well as 320,965 primary meQTLs and 614,195 independent meQTLs in the conditional analysis, revealing the comprehensive genetic architecture underlying methylation variation. The colocalization and mediation analyses from the present study also provide evidence supporting co-regulation of methylation and gene expression in African Americans. Overall, our results represent an important step toward revealing the co-regulation between DNA methylation and gene expression, facilitating the functional integration and interpretation of epigenetic and gene regulatory changes that influence human disease etiology in African Americans.

Comparing to the eQTL mapping study in the same cohort[26], we found that the proportion of CpG sites identified to harbor meQTLs (44%) is higher than the proportion of genes identified to harbor eQTLs (31.08%). The number of independent meQTLs per meCpG site (mean = 1.914; max = 19) is also higher than the number of independent eQTLs per eGene (mean = 1.474; max = 9). In addition, the identified meQTLs explain a slightly higher proportion of cis-SNP heritability than that by eQTLs. Specifically, the primary meQTLs explain a median

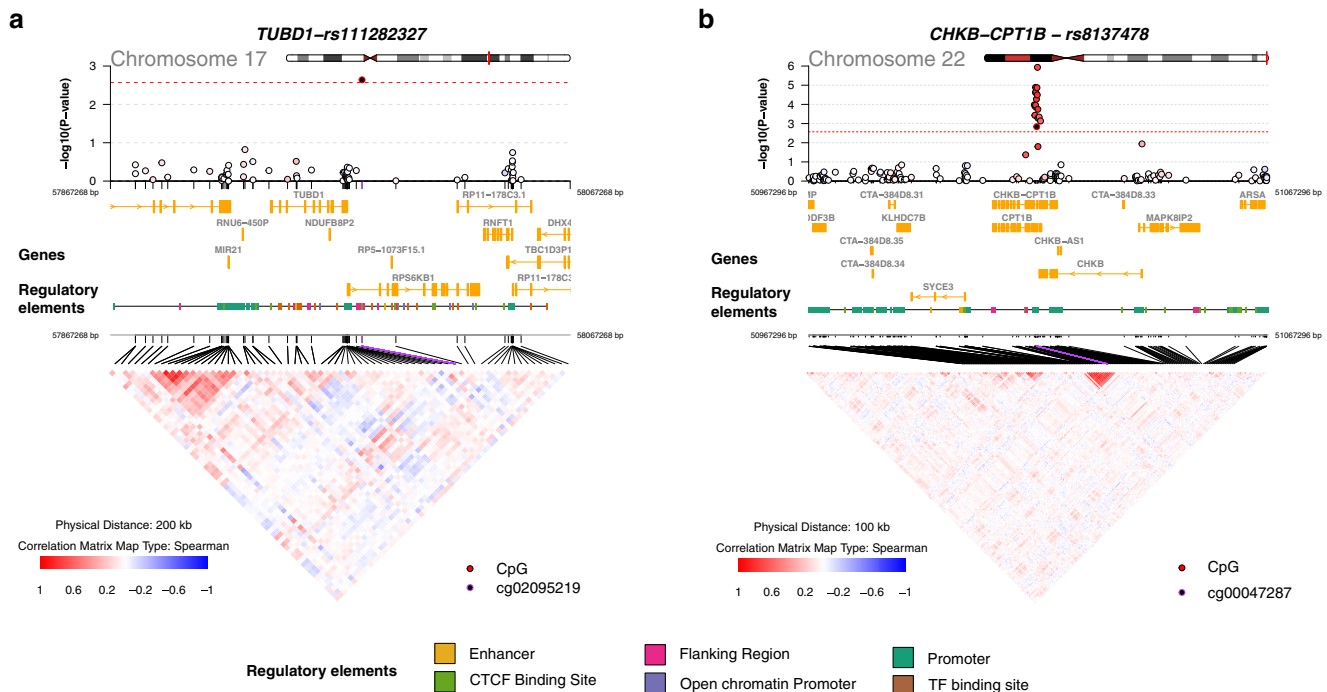

**Fig. 6 | Overlap with genomic annotations for co-localized eGene-meCpG pairs with strong evidence of mediation in the SME model. a** The *TUBD1* gene region with cg02095219 affected by a common causal variant, rs111282327. **b** The *CHKB*-*CPT1B* gene region with cg00047287 affected by a common causal variant, rs8137478. The *p*-values are obtained from two-sided Sobel tests. SME: SNP-Methylation-Expression direction.

of 68.62% cis-heritability of methylation while the primary eQTLs explain a median of 64.52% cis-heritability of expression. Both primary and conditional meQTLs explain a median of 87.35% cis-heritability of methylation while primary and conditional eQTLs explain a median of 77.83% cis-heritability of expression. These results suggest that methylation likely has stronger genetic determinant as compared to gene expression. Nevertheless, we acknowledge that the identified primary meQTLs do not explain all cis-SNP heritability in meCpGs, and neither do the additional conditional meQTLs identified through the conditional analysis. In addition, the cis-SNP heritability only represents a small proportion of total SNP heritability, suggesting that a large fraction of SNP heritability remains largely unidentified. Future studies with larger sample sizes are necessary to fully capture the polygenic genetic architecture underlying DNA methylation variation[20,24]. While the sample sizes of the eQTL study ($N = 1032$) and the meQTL study ($N = 961$) are similar, we acknowledge that many other factors may influence the comparison of results and contribute to the observed differences. First, the tissues for the samples in the expression and methylation data are different: the expression data are collected from lymphoblastoid cell lines (LCLs), while the methylation data are collected from peripheral blood leukocytes. Second, the technologies used for the measurement are different. The expression was measured using the Affymetrix Human Transcriptome Array 2.0, which surveys the entire transcriptome. In contrast, the methylation was assessed using the Illumina Infinium HumanMethylationEPIC BeadChip, which covers only 30% of the human methylome with a particular focus on regulatory regions[59]. The limited methylome coverage should be taken into consideration when generalizing the meQTL findings to the whole genome[38].

Comparing to the previous meQTL studies, we found a higher replication rate in African ancestry than other ancestries. For the PB and CBA samples from African ancestry, the replication rate is 0.98 in GENOA. In contrast, for the European ancestry samples from GoDMC, FHS, and Hawe et al., the replication rate in GENOA is 0.77, 0.82, and 0.93, respectively. In the South Asian ancestry

samples from BEST and Hawe et al., the replication rate is 0.91 and 0.93, respectively. This may be in part because the PB and CBA samples have a more similar ancestral background and LD structure than the FHS and BEST samples when compared with GENOA samples. Also, the FHS study has a much larger sample size, which allowed them to detect meQTLs with small effects that are not detectable by other studies. It also may be a direct result of the methylation platforms used, as the meQTL studies in African ancestry samples were conducted using the 27 K Bead-Chip, which is comprised almost exclusively of methylation sites in gene promoters and CpG islands[60]. The FHS and BEST meQTL studies were conducted using the 450 K array that interrogates a much broader array of CpG types across the epigenome. We also note that we did not perform tests on all of the significant SNP-CpG pairs obtained from other studies. This is because we selected different cis-window sizes for the SNP-CpG pairs and used different methylation platforms.

Multiple lines of future research are possible. First, multivariate analysis that jointly models multiple CpG sites or multiple SNPs, such as the recent high dimensional mediation analysis framework[61–63], could be beneficial as those methods can account for the correlations in methylation among CpG sites and the correlations among SNPs due to LD. Second, the methylation data in the present study are collected from peripheral blood leukocytes, which consist of a set of closely related cell types such as neutrophils, monocytes, eosinophils, basophils, and lymphocytes. Consequently, deconvolution of the methylation data in the future could enable cell-type specific meQTL mapping analysis for the identification of cell-type specific meQTLs. Third, extending the analysis beyond cis-meQTL mapping by conducting trans-meQTL mapping could help comprehensively characterize the genetic architecture underlying methylation[31]. Unfortunately, due to the extremely heavy computation burden resulted from the very large number of SNP-CpG pairs to be examined, we were unable to perform trans-meQTL mapping in the present study (Supplementary Table 9). Finally, collecting methylation data from

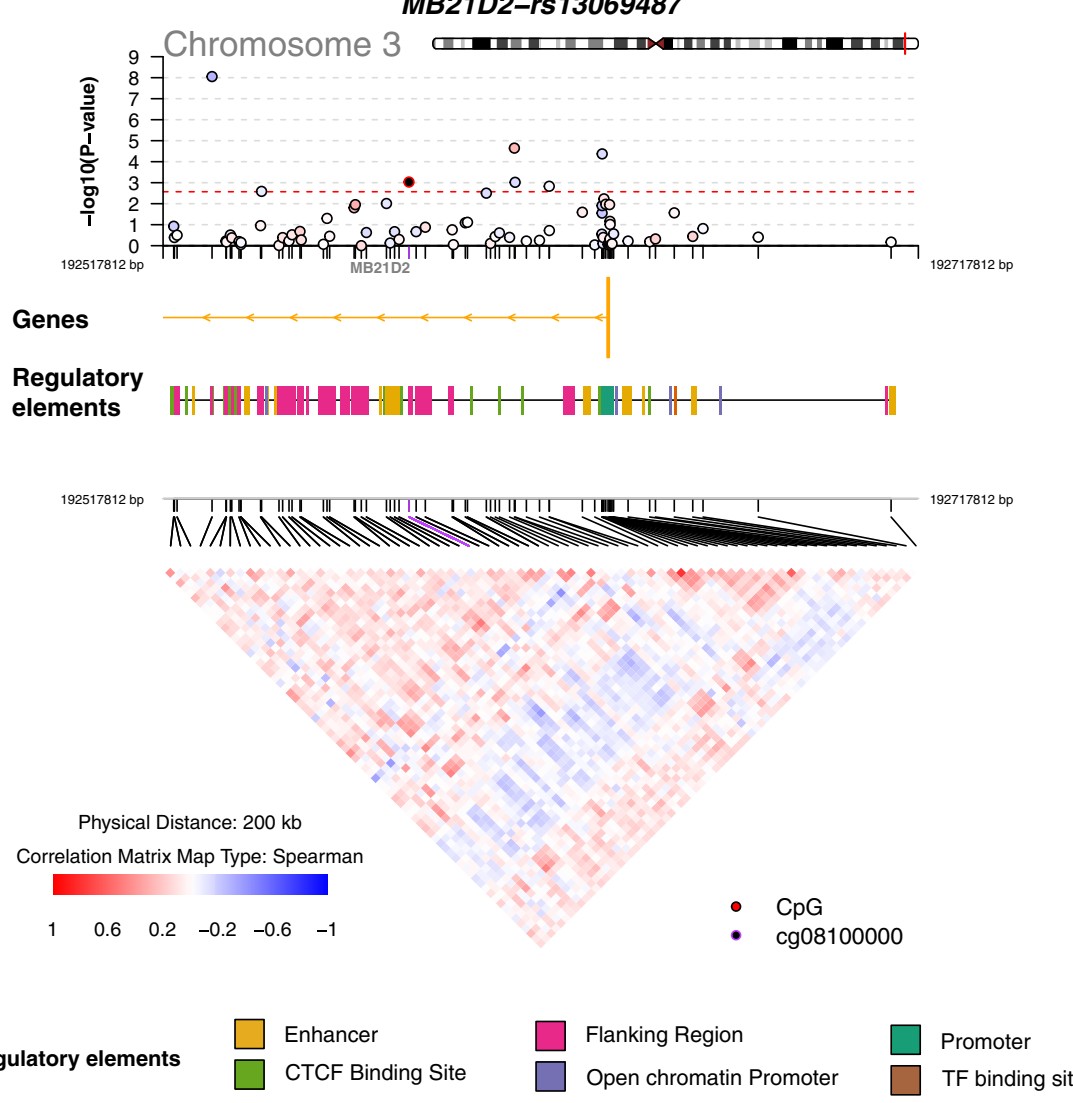

**Fig. 7 | Overlap with genomic annotations for co-localized eGene-meCpG pair with strong evidence of mediation in the SEM model.** The *MB21D2* gene region with cg08100000 affected by a common causal variant, rs13069487. The *p*-values are obtained from two-sided Sobel tests. SEM: SNP-Expression-Methylation direction.

populations with additional ancestries can help us better understand the diverse genetic architectures underlying methylation variation in the different populations.

## Methods
### GENOA study
The Genetic Epidemiology Network of Arteriopathy (GENOA) study is a community-based study of hypertensive sibships that was designed to investigate the genetics of hypertension and target organ damage. The study includes African Americans (AA) from Jackson. In the initial phase of GENOA (Phase I: 1996–2001), all members of sibships containing at least 2 individuals with essential hypertension clinically diagnosed before age 60 were invited to participate, including both hypertensive and normotensive siblings. The exclusion criteria for GENOA included secondary hypertension, alcoholism or drug abuse, pregnancy, insulin-dependent diabetes mellitus, or active malignancy. Eighty percent of AA ($N = 1482$) from the initial study population returned for the second examination (Phase II: 2001–2005). Demographic information, medical history, clinical characteristics, lifestyle factors, and blood samples were collected in each phase. Written informed consent was obtained from all subjects and approval was

granted by participating institutional review boards (University of Michigan, University of Mississippi Medical Center, and Mayo Clinic).

### Genotype data and quality control
AA blood samples were genotyped using either the Affymetrix Genome-wide Human SNP Array 6.0 platform or the Illumina Human1M-DUO BeadChip. For each platform, participants were excluded if they had an overall SNP call rate <95% or sex mismatch between genotype and self-report. SNPs were excluded if they had a call rate <95%. Principal component analysis was performed to identify and remove samples whose genotype profile appeared to be different from all other samples (outliers). After removing outliers, there were 1599 AA samples with available genotype data. Imputation was performed using the Segmented HAPlotype Estimation & Imputation Tool (SHAPEIT[64]), version v2.r and IMPUTE version 2[65] using the 1000 Genomes project Phase I integrated variant set release (v3) in NCBI build 37 (hg19) coordinates (released on March 2012). Since genotyping was performed on multiple platforms, imputation was performed separately by platform and then the imputed data were combined. After imputation, SNPs with minor allele frequency (MAF) ≤ 0.01 or imputation quality score (info score) ≤ 0.4 in any

platform-based imputation were removed, leaving 30,022,375 markers covering both SNPs and SNVs/INDELs. After removing the SNVs/INDELs with PLINK[66], we retained 28,681,763 SNPs. The GENESIS package in R was used to infer population structure[67]. We used the PC-AiR function to extract the first five genotype PCs and used GEMMA[68] to estimate an individual relatedness matrix. We controlled for both PCs and the relatedness matrix in the meQTL mapping analysis.

## Methylation data and quality control

Genomic DNA was extracted from stored peripheral blood leukocytes that was collected at Phase 1 ($N = 1106$) or Phase 2 ($N = 304$) using AutoGen FlexStar (AutoGen, Holliston, MA). Bisulfite conversion was performed with the EZ DNA Methylation Kit (Zymo Research, Irvine, CA), and methylation was assessed using the Illumina Infinium HumanMethylationEPIC BeadChip. The shinyMethyl R package was used to generate the density plot of the raw intensity data to identify sex mismatches or sample outliers. Samples with incomplete bisulfite conversion were identified and removed using the QCinfo() function in the ENmix R package. Sample identity was checked using the 59 SNP probes implemented in the EPIC chip, and mismatched samples were removed. Next, the Minfi R package was used to perform background correction and normalization using Noob. The regression on correlated probes (RCP) method was used to adjust probe-type bias. Individual probes with detection $p$-value <1e−16 were considered successfully detected. Samples and probes with detection rate <10% were removed. Following these steps, 857,121 probes for 1100 samples at Phase I and 294 samples at Phase II remained for analysis. Next, we removed CpG sites on the X and Y chromosomes. We then followed[15] to further remove 36,860 cross-reactive probes[59] and 24,495 probes with a SNP at the target CpG site or within a single-base extension, resulting in a final set of 771,134 CpG sites. White blood cell counts were estimated using Houseman method. The extracted methylation data were then converted from β-values to M-values[69] through $\log_2 \frac{\beta}{1-\beta}$. Here the methylation β-values are often interpreted as the proportion of methylation at a given site. Converting β-values to M-values better controls for the heteroskedasticity that is present in the β-values. In this analysis, M-values are treated as the outcome variable. We used a linear mixed model to remove batch effects, extracted residuals from the model, and quantile normalized the residuals across CpG sites for the meQTL analysis.

## meQTL mapping analysis

The meQTL mapping analysis was performed using individuals with both genotype and methylation data ($N = 961$). For each CpG site in turn, we followed[18,21] to extract cis-SNPs that are +/−50 kb of the CpG site. We performed a sensitivity analysis on chromosome 22 using cis-SNPs that reside +/−50 kb, +/−500 kb, or +/−1 Mb of the CpG site for cis-meQTL mapping and heritability estimation and found similar results across different window sizes; thus, we used +/−50 kb for subsequent analyses. In the meQTL mapping analysis, we focused on a set of 728,578 CpG sites that contain at least one cis-SNP and genotype information for 8,993,056 imputed cis-SNPs on 961 AA individuals that also have methylation data. The median number of cis-SNPs per analyzed CpG site is 300 (mean = 304.1; sd = 136.3), with range varying from 1 to 2298. For each analyzed CpG site, we examined one cis-SNP at a time and applied a linear mixed model implemented in GEMMA[68] for meQTL mapping.

For each variant, we fitted the following linear mixed model:

$$\mathbf{y} = \mu + \mathbf{x}\beta + \mathbf{u} + \mathbf{e}, \tag{1}$$

$$\mathbf{u} \sim MVN(0, \sigma_u^2 \mathbf{K}), \tag{2}$$

$$\mathbf{e} \sim MVN(0, \sigma_e^2 \mathbf{I}), \tag{3}$$

and tested the null hypothesis H$_0$: $\beta = 0$ vs the alternative H$_1$: $\beta \neq 0$. Above, $\mathbf{y}$ is the $n$ by 1 vector of residual DNA methylation levels at a CpG site for the $n$ individuals, where the residual DNA methylation levels were obtained by first correcting for age, gender, cell type proportions of five cell types (including CD8 T cells, CD4 T cells, NK cells, B cell, and monocyte cells), the top five genotype PCs from PC-AiR; $\mu$ is the intercept; $\mathbf{x}$ is the $n$ by 1 vector of genotypes for the genetic variant of interest; $\beta$ is the variant's effect size; the $n$ by 1 vector of $\mathbf{u}$ is a random effects term to control for individual relatedness and other sources of population structure, where the $n$ by $n$ matrix $\mathbf{K}$ is the genetic relatedness matrix calculated from GEMMA; the residual errors are represented by $\mathbf{e}$, an $n$ by 1 vector, and MVN denotes the multivariate normal distribution. We used the command "gemma -bfile cpgname -maf 0.01 -r2 0.9999999 -hwe 0 -n phenotype -k relatedness -lmm 1 -o output" to fit the model in GEMMA.

After mixed model analysis, we selected the SNP with the lowest $p$-value for each CpG site as the candidate meQTL and used its $p$-value as the CpG site-level significance measure. We permuted the sample labels 10 times and applied the same meQTL mapping procedure to obtain an empirical null distribution of CpG site-level $p$-values[70–72]. As a sensitivity analysis, we also evaluated constructing the empirical significance cutoff using 2, 5, 10, 50, 100, 200, 300, 400, and 500 permutations on chromosome 22; however, since the results were similar, we used 10 permutations for our subsequent analyses. With the empirical null distribution, we computed the false discovery rate (FDR) associated with each $p$-value threshold following refs. [70,71] and obtained the $p$-value threshold that corresponded to a 5% FDR control, which is 2.267195e−4 in the present study. We refer to any CpG site that passes the FDR threshold of 5% as an meCpG and refer to the SNP with the lowest $p$-value in each meCpG site as the primary meQTL. Following refs. [70,71], we also refer to any cis-SNP with a significant association with a meCpG site as an meQTL.

For each meCpG in turn, we performed conditional analysis using forward and backward stepwise selection to identify additional conditional meQTLs following refs. [72,73]. To do so, we refer to the primary meQTLs as E1 SNPs. In the forward stages, for each CpG in turn, we performed association analysis conditional on the E1 SNP and identified the strongest SNP association among the remaining SNPs. We refer to the identified SNP as an E2 SNP if its conditional $p$-value is below the genome-wide significance threshold established above (2.267195e−4). Next, we performed further association analysis conditional on both E1 and E2 SNPs to identify E3 SNPs. We repeated this process until the smallest p-value among the remaining SNPs no longer exceeded the genome-wide significance threshold. In the backward stages, we tested each variant that is already included in the model while controlling for all the other included variants for a CpG site and dropped the variant if it was not significant.

## Comparison with eQTL mapping results

We obtained eQTL mapping results on GENOA AA samples from a previous study[26]. The previous eQTL mapping analysis focused on expression measurements for 17,616 protein-coding genes and genotype information for 30,022,375 imputed SNPs from 1032 AA individuals. Among the 1032 AA individuals used in eQTL mapping, 800 overlapped with the 961 individuals used in the present meQTL mapping analysis. In the previous eQTL analysis, the gene expression data were processed with Combat[74] to remove batch effects or other technical covariates. The cis-SNPs within 100 kb of each gene were extracted and the linear mixed models implemented in GEMMA[68] were used for eQTL mapping. The eQTL mapping adjusted for age, gender, the top five genetic principal components (PCs), as well as a genetic

relatedness matrix to control for familial relationships. We identified 5475 primary eQTLs and used them in the colocalization analysis described in the following section.

## Comparison of cis-meQTL results with previous studies

We compared our cis-meQTL mapping results to six previous cis-meQTL mapping studies that include the Genetics of DNA Methylation Consortium (GoDMC)[24], Hawe et al.[25], Framingham Heart Study (FHS)[20], a study on adult peripheral blood (PB) samples, and umbilical cord blood at birth (CBA) samples[21,75,76], and the Bangladesh Vitamin E and Selenium Trial (BEST) study[15]. We selected these cis-meQTL studies for comparison because they are blood-based genome-wide meQTL studies with sample size greater than 100 from diverse ancestries. We did not compare with the HapMap meQTL study[18] due to its small sample size and low detection power for cis-meQTLs.

The GoDMC study[24] measured DNA methylation levels in whole blood using the Illumina HumanMethylation 450 K BeadChip. Cis-meQTL mapping analysis in the GoDMC study was performed on 27,750 European participants, with 420,509 CpG sites and 10 million SNPs. Among the 71,616,458 cis-SNP-CpG pairs separated by <1 Mb, the GoDMC study identified 169,370 significant meCpGs with $p$-values below 1e−8. The GoDMC study made SNP-CpG pairs with nominal significant $p$-values in at least one study publicly available. In the comparison, we matched CpG sites between the two studies using CpG site IDs and matched SNPs using base pair positions. Among the 71,616,458 SNP-CpG pairs available in European population, the SNPs and CpG sites are separated by <50 kb in 21,023,205 pairs, and 9,010,077 of them were tested in GENOA AA samples.

Hawe et al.[25] measured DNA methylation levels in peripheral blood using the Illumina HumanMethylation 450 K BeadChip. Cis-meQTL mapping analysis in Hawe et al. was performed on 3799 European and 3195 South Asian participants. Among the cis-SNP-CpG pairs separated by <1 Mb, Hawe et al. identified 41,608 significant meCpGs in Europeans and 69,495 significant meCpGs in South Asians with p-values below 1e−14. The Hawe et al. study made SNP-CpG pairs with significant p-values in at least one study publicly available. In the comparison, we matched CpG sites between the two studies using CpG site IDs and matched SNPs using base pair positions. Among the 11,165,559 SNP-CpG pairs available in European and South Asian population, the SNPs and CpG sites are separated by <50 kb in 2,507,048 (European) and 4,292,518 (South Asian) pairs, 1,338,027 and 1,223,921 of them were tested in GENOA AA samples.

The FHS study[20] measured DNA methylation levels using the Illumina Infinium Human Methylation450 BeadChip (450 K) in the whole blood buffy coat samples for 4170 European ancestry participants. The cis-meQTL mapping analysis in FHS was performed using 415,318 CpG sites and 8.5 million SNPs. SNPs and CpG sites separated by <1 Mb were tested for association. This study identified 4,447,327 unique meQTLs at the Bonferroni-corrected $p$-value threshold of 0.05, along with 121,599 meCpGs and 27,235,697 significant SNP-CpG pairs. The FHS study only made available the significant SNP-CpG pairs at the Bonferroni-corrected $p$-value threshold of 0.05. We obtained the significant SNP-CpG pairs from FHS and compared them with our meQTL mapping results. In the comparison, we matched CpG sites between the two studies using CpG site IDs and matched SNPs using base pair positions. Among the 27,235,697 significant SNP-CpG pairs available in the FHS study, the SNPs and CpG sites are separated by <50 kb in 10,894,656 pairs, and 4,951,955 of them were tested in GENOA AA samples.

The meQTL mapping studies on PB[76] and CBA samples[75] ($n = 177$ in total) were performed on individuals with African ancestry, with detailed analysis in ref. 21. Briefly, the PB study contains $n = 90$ samples and the CBA study contains $n = 87$ samples, with a common set of 222,888 SNPs and 20,093 CpG sites measured in both studies using the HumanMethylation27 BeadChip (Illumina). The cis-meQTL mapping analyses in both studies were performed on 529,224 unique SNP-CpG pairs that were separated by <50 kb. The cis-meQTL mapping analysis detected 724 cis-meQTLs in PB and 629 cis-meQTLs in CBA with a conservative Holm adjustment for multiple testing, with these significant SNP-CpG pairs publicly available. We obtained the significant SNP-CpG pairs from both studies and compared them with our results. In the comparison, we matched CpG sites between the two studies using the CpG site IDs and matched SNPs using base pair positions. Among the 724 significant SNP-CpG pairs available in PB, 517 of them were tested in GENOA AA samples. Among the 629 significant SNP-CpG pairs available in CBA, 531 of them were tested in GENOA AA samples.

Finally, the BEST study[15] measured DNA methylation levels in whole blood using Illumina HumanMethylation 450 K BeadChip kit (Illumina). Cis-meQTL mapping analysis in the BEST study was performed on 337 Bangladesh samples, with 423,604 CpG sites and 8,639,940 SNPs. Among the 994,862,964 SNP-CpG pairs separated by <500 kb, the BEST study identified 77,664 significant meCpGs with empirical $p$-values below the FDR threshold of 0.01. The BEST study made all SNP-CpG pairs with $p$-value <0.05 publicly available. We obtained the SNP-CpG pairs, ranked the meCpGs based on the smallest $p$-value in each meCpG, extracted the top 10,507,576 significant SNP-CpG pairs that correspond to the 77,664 significant meCpGs, and compared them with our results. In the comparison, we matched CpG sites between the two studies using CpG site IDs and matched SNPs using base pair positions. Among the 94,168,979 SNP-CpG pairs available in the BEST study, the SNPs and CpG sites are separated by <50 kb in 17,862,650 pairs, and 8,395,396 of them were tested in GENOA AA samples.

We compared the meQTL mapping results from each of the above studies. In each comparison, we calculated the replication rate (Storey's $\pi_1$), which is defined as the expected true positive rate and was estimated by selecting significant SNP-CpG pairs in each comparison cohort and examining their $p$ value distribution in GENOA[77,78]. Therefore, the replication rate effectively captures the proportion of signals in the previous study that is replicated in the present study. We were unable to calculate $\pi_1$ in the reversed way to estimate the proportion of signals in the present study that is replicated in the previous study, because the previous meQTL studies often analyzed a different set of SNP-CpG pairs and reported only the significant SNP-CpG pairs but not all the tested pairs. Specifically, the FHS (EU) study only provided significant SNP-CpG pairs with $p$-value <2e−11. The Hawe et al. 2022 (EU and SA) only provided SNP-CpG pairs significant ($p < 1e−14$) in at least one population. The GoDMC (EU) only provided SNP-CpG pairs with $p$-value <1e−5 in at least one cohort and provided association statistics from meta-analysis. The BEST study (SA) only provided SNP-CpG pairs with $p$-value <0.05, and did not provide MAF or allele information. The PB and CBA studies (African) only provided significant pairs with $p$-value <0.05 after Holm adjustment. Consequently, when some meQTLs identified in the present study were not replicated in the previous studies, we often do not know whether the previous studies did not identify these meQTLs or simply did not test these SNP-CpG pairs.

Because previous studies used different significance thresholds, we examined three different thresholds in the comparison: (1) the $p$-value threshold used in the original study, (2) a $p$-value threshold of 1e−5 for both studies, and (3) the Bonferroni corrected $p$-value threshold of 0.05 for the tested SNP-CpG pairs in both studies. In each comparison, we also separated the meQTLs from the present study into two sets: the ones that replicate previous study and the ones that do not. We then compared SNP effect sizes and allele frequencies between the two sets.

## Methylation heritability estimation and partitioning

For each CpG site, we estimated the proportion of variance in methylation level explained by all SNPs using the Bayesian sparse

linear mixed model (BSLMM) implemented in GEMMA. Following ref. 79, we also used BSLMM[80] to partition the methylation level variance into a cis-component and a trans-component. The cis-component represents the proportion of methylation level variance explained by cis-SNPs while a trans-component represents the proportion of methylation level variance explained by trans-SNPs. The cis-SNPs were defined as those SNPs that reside within 50 kb of the CpG site as described above while the trans-SNPs were defined as all other SNPs.

## Functional enrichment of meCpG sites

We examined whether the identified meCpG sites are enriched in functional genomic regions defined based on four sets of genomic annotations. The first set of genomic annotations are obtained from the R package "IlluminaHumanMethylationEPI-Canno.ilm10b2.hg19". In particular, the R package annotated the analyzed CpG sites based on their relation to CpG islands, which are short genomic regions with high CpG density, into the following six regions: CpG islands, north and south shores, north and south shelves, and open sea[81]. Specifically, CpG shores are defined as the 2 kb of sequence flanking of a CpG island. CpG shelves are defined as the 2 kb of sequence further flanking CpG shores. While the remaining regions outside of CpG island/shore/shelf are denoted as open sea[81].

The second set of genomic annotations are obtained based on the relative position of CpG sites with respect to genes. It includes seven genomic annotations: TSS1500 (if the CpG site resides 200–1500 bases upstream of the transcription start site, TSS, of a gene), TSS200 (if the CpG site resides 0–200 bases upstream of the TSS of a gene), 5′UTR (if the CpG site resides within the 5′ untranslated region of a gene, between its TSS and the ATG start site), 1st exon (if the CpG site resides in the first exon of a gene), gene body (if the CpG site resides between the ATG and the stop codon), 3′UTR (if the CpG site resides between the stop codon and poly A tail), and intergenic regions[81]. Following[82], if a CpG site has multiple annotations, we select one according to the following criteria: TSS200 > TSS1500 > 5′UTR > 1st Exon > Body > 3′ UTR > Intergenic.

The third set of genomic annotations are based on the epigenetic states inferred from ChromHMM using the functional genomic data from Roadmap reference epigenomes[83] and ENCODE[84,85]. Specifically, we downloaded 111 Roadmap reference epigenomes[83] and 16 additional epigenomes from ENCODE[84,85] for a wide range of human cell types and tissue types. We focused on 25 blood-related epigenomes (E062, E034, E045, E033, E044, E043, E039, E041, E042, E040, E037, E048, E038, E047, E029, E031, E035, E051, E050, E036, E032, E046, E030, E116, E124) and used the ChromHMM model to define 15 epigenetic states: (1) active TSS (TssA), (2) flanking active TSS (TssAFlnk), (3) transcription at gene 5′ and 3′ (TxFlnk), (4) strong transcription (Tx), (5) weak transcription (TxWk), (6) genic enhancers (EnhG), (7) enhancers (Enh), (8) zinc finger genes and repeats (ZNF/Rpts), (9) hetero-chromatin (Het), (10) bivalent/poised TSS (TssBiv), (11) flanking bivalent TSS/enhancers (BivFlnk) (12) bivalent enhancers (EnhBiv), (13) repressed Polycomb (ReprPC), (14) weak repressed Polycomb (ReprPCWk), and (15) quiescent/low (Quies).

The last set of genomic annotations are based on the positions of regulatory motifs. Specifically, we downloaded 3,961,042 predicted human regulatory motif sites (hg19) for 607 motifs from the motifmap[86] (http://motifmap.ics.uci.edu) and annotated 79,331 SNPs in the present study to be within the motif binding sites.

For each of the above functional genomic or motif annotations in turn, we performed Fisher's exact test to examine whether the meQTLs are enriched with the annotation. In addition, we performed enrichment analysis using Torus[37], which models the meQTL signal landscape and multiple annotations jointly.

## Co-localization analysis

We obtained a list of eGene-meCpG pairs for co-localization analysis to examine whether a common causal variant may influence both gene expression and methylation (Supplementary Fig. 1). To do so, we first obtained 5475 primary eQTLs from GENOA AA samples based on the eQTL mapping analysis. We also obtained 35,629 meCpG sites that are significantly associated with the 5475 primary eQTLs at the 5% FDR threshold in the meQTL mapping analysis. Because some of the meCpG sites are correlated with each other and influenced by the same cis-SNP[87], we followed refs. 15,88 and pruned the list of meCpG sites. Specifically, if a primary eQTL is associated with multiple meCpG sites, we only retained the meCpG site whose primary meQTL had the highest LD with the primary eQTL. We then extracted the retained meCpG site, paired it with the eGene, with the eQTL being the link, to obtain a total of 4854 eGene-meCpG pairs. The 4854 eGene-meCpG pairs were used for down-stream co-localization and mediation analyses.

We performed co-localization analysis using coloc[39] on each eGene-meCpG pair to examine the extent to which a single causal variant affects both gene expression and methylation. For each eGene-meCpG pair, we obtained two sets of summary statistics in the form of p-values to serve as inputs for coloc: one set from the eQTL analysis representing the association evidence between SNPs and gene expression, and the other set from the meQTL analysis representing the association evidence between SNPs and methylation. Specifically, for each pair of eGene-meCpG, we obtained the association results for all SNPs within +/−50 kb of the eQTL from the eQTL analysis and within +/−50 kb of the meQTL from the meQTL analysis. Colocalization analysis in coloc requires specifying prior probabilities that a SNP is associated with gene expression ($p_1$), methylation ($p_2$), or both ($p_{12}$). We set $p_1 + p_{12}$ at 0.00084 as we identified 5406 unique primary eQTLs among the 6,432,684 examined cis-SNPs in the eQTL analysis. We set $p_2 + p_{12}$ to be 0.03 as we identified 254,113 unique primary meQTLs among 8,993,056 cis-SNPs in the meQTL analysis. Following the recommendation of ref. 15, we examined six choices of $p_{12}$ (4.2e−05, 8.4e−05, 2.1e−04, 4.2e−04, 6.3e−04, 7.56e−04), corresponding to the probability that an eQTL is also an meQTL being either 5%, 10%, 25%, 50%, 75%, or 90%. Also following ref. 15, we evaluated the validity of the six $p_{12}$ choices using internal empirical calibration[89], which assesses the similarity between the prior and posterior expectations from the colocalization analysis. With this criterion, we selected 6.3e−04 as the best choice for $p_{12}$ (Supplementary Fig. 21), which corresponds to a prior probability of 75% that an eQTL is also an meQTL and a prior probability of 2% that an meQTL is also an eQTL. With these prior choices, we performed colocalization analysis and declared the eGene-meCpG pair to share a colocalized SNP if the posterior probability of sharing, PP4, is greater than 0.8 as recommended by coloc.

For each GWAS trait in turn, we performed colocalization analysis using both the default setting of coloc and the Susie version of coloc to estimate the posterior probability that the same variant is responsible for GWAS-meQTL or for GWAS-eQTL. Specifically, among the eGene-meCpG pairs used in co-localization analysis in the previous paragraph, we extracted eQTL and meQTL association results for SNPs that reside within 50 kb of the lead-eQTL and intersected these SNPs with the GWAS SNPs based on SNP positions to arrive at a common set of SNPs. Only pairs with at least one GWAS association signal that surpassed a relaxed significance threshold ($p < 1e−5$) were included in the analysis. In addition, for the GWAS-meQTL colocalization analysis, in addition to using only the CpGs in the eGene-meCpG pairs, we also performed a genome-wide scanning of GWAS-meQTL colocalization signals. To do so, we followed[36] and split the GWAS summary statistics into 2583 approximately linkage disequilibrium (LD) independent regions[90]. We only included GWAS hit loci that encompassed a GWAS significant ($p < 5e−8$ and $p < 1e−5$) signal and with at least 50 SNPs in common between the SNPs in GWAS and SNPs tested in an meCpG.

In addition, we performed colocalization analysis using a matched number of European samples from the UK Biobank (UKBB) for SBP, DBP, and BMI, as we did not have access to HTN, PP, and the phecode of T2D in our UKBB application. For each of the three traits in turn, we obtained a random subsample of individuals of white British ancestry to match the sample size of the African ancestry GWAS studies. We then performed GWAS analysis where we included age, sex, and top 20 genetic PCs as covariates to obtain summary statistics. Afterwards, we performed colocalization analysis with coloc (between GWAS and meQTLs, and between GWAS and eQTLs) and performed multi-traits colocalization analysis with moloc (among GWAS, meQTLs and eQTLs), using the GWAS summary statistics from UKBB.

## Multi-trait co-localization analysis

We obtained GWAS summary statistics for six complex traits that include type 2 diabetes (T2D), body mass index (BMI), and four blood pressure traits including pulse pressure (PP), diastolic blood pressure (DBP), systolic blood pressure (SBP), and hypertension (HTN). All GWAS were conducted in African ancestry samples. For T2D, we obtained the GWAS summary statistics from the MEta-analysis of type 2 DIabetes in African Americans (MEDIA) Consortium study, which is a meta-analysis of 17 GWASs with a total of 8284 cases and 15,543 controls[40]. For BMI, we obtained the GWAS summary statistics from the African Ancestry Anthropometry Genetics Consortium (AAAGC) with 52,895 individuals[41]. For the four blood pressure traits, we obtained the GWAS summary statistics from COGENT-BP meta-analyses with a total of 31,968 individuals from 21 African ancestry cohorts[42].

For each GWAS trait in turn, we performed multi-trait colocalization analysis using moloc[91] to estimate the posterior probability that the same variant is shared across GWAS traits, eGenes and meCpGs. To do so, in addition to the eQTL and meQTL summary statistics from the GENOA study that were used in the coloc analysis, we included the additional GWAS summary statistics in the moloc analysis. Specifically, on each eGene-meCpG pair, we extracted association results for all SNPs that reside within 100 kb of the eQTL from the eQTL and meQTL analyses. We intersected these SNPs with the GWAS SNPs based on SNP IDs to arrive at a common set of SNPs shared between datasets. We retained eGene-meCpG pairs with association results for at least 10 SNPs in the moloc analysis. In moloc, 15 configurations of possible variant sharing schemes were calculated across the GWAS trait, gene expression, and methylation[91]. We used the posterior probability of association (PPAs) threshold of 0.8 as evidence for multi-trait co-localization. We followed moloc recommendation[44,91] and set the prior probabilities to be 1e−4, 1e−6, and 1e−7 for the association of one, two, or three traits, respectively.

## Mediation analysis

We performed mediation analysis on the eGene-meCpG pairs used in the co-localization analysis using 800 individuals who have gene expression, methylation, and genotype data. We conducted the mediation analysis in two directions. The first is the SNP-Methylation-Expression (SME) direction, where the SNP genotype is treated as the exposure, methylation level for the CpG site is treated as the mediator, and gene expression is treated as the outcome. The second is the SNP-Expression-Methylation (SEM) direction, where the SNP genotype is treated as the exposure, gene expression is treated as the mediator, and methylation is treated as the outcome. In both types of mediation analyses, for methylation, we adjusted for gender, age, top 5 genetics PCs, white blood cell type proportions, and batch effects using the lmer function in R, and then took the residuals and quantile normalized them. For gene expression, we used ComBat method in the sva R package[74] to remove batch effects, and adjusted for age, gender, the top five genotype PCs, and then took the residuals and quantile normalized them. We then re-estimate the exposure-outcome association

adjusting for the mediator in the following regression:

$$\mathbf{Y} = \mu_1 + \beta_3 \mathbf{X} + \beta_1 \mathbf{M} + \epsilon_1, \quad (4)$$

where $\mathbf{X}$ is the exposure, $\mathbf{M}$ is the mediator, $\mathbf{Y}$ is the outcome, $\mu_1$ is the intercept, $\beta_3$ is the effect of exposure on outcome, $\beta_1$ is the effect of mediator on outcome, and $\epsilon_1$ is the residual error.

We also estimated the exposure on mediator effect through the following regression:

$$\mathbf{M} = \mu_2 + \beta_2 \mathbf{X} + \epsilon_2, \quad (5)$$

where $\mu_2$ is the intercept, $\beta_2$ is the effect of exposure on mediator, and $\epsilon_2$ is the residual error.

We then obtained the Sobel test statistics for testing the mediation effect in the form of

$$t = \frac{\beta_1 \beta_2}{\sqrt{\beta_1^2 \sigma_{\beta_2}^2 + \beta_2^2 \sigma_{\beta_1}^2}}, \quad (6)$$

where $\sigma_{\beta_1}^2$ and $\sigma_{\beta_2}^2$ are the variance of $\beta_1$ and $\beta_2$. $\beta_1 \beta_2$ is often referred as the indirect effect or mediation effect. With the Sobel test statistics, we obtained the corresponding $p$-values based on a standard normal distribution. We declared significance at an FDR of 0.05.

Following[92], we also calculated the mediation proportion, which represents the proportion of the total effect of the exposure on the outcome mediated through the mediator, in the form of:

$$\rho = \frac{\beta_1 \beta_2}{\beta_3 + \beta_1 \beta_2}, \quad (7)$$

where $\rho$ is the mediation proportion, $\beta_1 \beta_2$ is the indirect effect of exposure on outcome mediated through the mediator.

## Reporting summary

Further information on research design is available in the Nature Portfolio Reporting Summary linked to this article.

## Data availability

The methylation data generated in this study have been deposited in the Gene Expression Omnibus (GEO) database under accession code GSE210256. The genotype data are available in the Database of Genotypes and Phenotypes (dbGaP) under accession number phs001238.v2.p1 under restricted access due to the requirement of Institutional Review Board (IRB), access can be obtained by written request to J.A.S. (smjenn@umich.edu) and S.L.R.K. (skardia@umich.edu) who will aim to respond to requests within 2 weeks. The UK Biobank data are from UK Biobank resource under application number 30686. The human regulatory motif sites are downloaded from http://motifmap.ics.uci.edu[86]. The summary statistics from the GoDMC study[24] are available at http://mqtldb.godmc.org.uk. The summary statistics from Hawe et al.[25] are available at https://zenodo.org/record/5196216#.YRZ3TfJxeUk. The summary statistics from the FHS study[20] are available at https://ftp.ncbi.nlm.nih.gov/eqtl/original_submissions/FHS_meQTLs. The summary statistics from the PB and CBA samples[21] are available at https://static-content.springer.com/esm/art%3A10.1186%2F1471-2164-15-145/MediaObjects/12864_2013_5906_MOESM1_ESM.csv. The summary statistics from the BEST study[15] are available at https://datadryad.org/stash/dataset/doi:10.5061/dryad.hq68q. The summary statistics from the GENOA eQTL mapping analysis[26] are available at http://www.xzlab.org/data.html. The gene expression data used in the GENOA eQTL mapping analysis is available at the Gene Expression Omnibus (GEO) database under accession codes GSE138914 for AA and GSE49531 for EA. The summary statistics (mapped to Genome

Assembly GRCh37) generated in the GENOA meQTL mapping analysis are available at http://www.xzlab.org/data.html and https://doi.org/10.5281/zenodo.7697509[93]. The authors declare that all the other data supporting the findings of this study are available within the article, its Supplementary Information file or from the corresponding author upon reasonable request.

## Code availability

The code used to reproduce the analysis in this study is available at Github repository https://github.com/shangll123/GENOA_meQTL and at Zenodo platform (https://doi.org/10.5281/zenodo.7697509)[93].

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

## Acknowledgements
Support for the Genetic Epidemiology Network of Arteriopathy (GENOA) data collection and analysis was provided by the National Heart, Lung and Blood Institute (U01HL054457 to S.L.R.K., RC1HL100185 to S.L.R.K., R01HL119443 to S.L.R.K., R01HL133221 to J.A.S., and R01HL141292 to J.A.S.) and the National Institute of Neurological Disorders and Stroke (R01NS041558 to Stephen T. Turner) of the National Institutes of Health. This study was also partially supported by the National Human Genome Research Institutes grant R01HG009124 to X.Z. The funders had no role in study design, data collection and analysis, decision to publish, or preparation of the manuscript.

## Author contributions
X.Z. and J.A.S. conceived and designed the study. L.S. performed statistical analysis with input from X.Z. and J.A.S. L.S., X.Z. and J.A.S. wrote the manuscript with input from W.Z., Y.Z.W., Z.L., J.J.C., M.K., T.H.M., and S.L.R.K. All authors read and approved the final manuscript.

## Competing interests
The authors declare no competing interests.

## Informed consent
Written informed consent was obtained from all participants, and Institutional Review Boards at the University of Michigan, University of Mississippi Medical Center, and Mayo Clinic approved this study.
