## [Peer Review File · Nature Communications]

meQTL Mapping in GENOA Reveals Genetic Determinants of DNA Methylation in African AmericansREVIEWER COMMENTS

Reviewer #1 (Remarks to the Author):

This study reports the mapping of DNA methylation QTLs (mQTLs) in blood cells for a sample of 961 African American individuals. Analyses and approaches are sound. The primary novelty of this work is the focus of individuals of African ancestry, a group underrepresented in prior studies. The paper would benefit from an increased focus on the knowledge gained by specifically from a focus on AA for QTL mapping (e.g., for QTL discovery, coloc, mechanisms). The authors should consider more clearly defining which scientific insights are novel, with clearer comparisons with results from prior studies. The results are a bit too descriptive, and they needs more context and biological interpretation. Specific comments are below;

Abstract

--line 54-56. This finding is well-known, so should be noted here as consistent with prior studies.

--line 60. overlap or colocate? Pairwise colocalization analysis goes beyond overlap as it aims to detect whether a causal variant is shared.

--line 63. Confusing sentence: what is the relation between mediation analysis and heritability?

--line 64. SNP or variant (including small indels)?

Introduction:

--the authors appear to assume that DNA methylation mediates the effect of SNPs on gene expression. Its is also possible that the epigenome responds to changes in gene expression/activity.

--the authors should acknowledge that DNA methylation represents chromatin conformation/structure and related histone modifications.

Methods

--line 161. The number of probes that passed QC seems on the high end. Authors should consider removing probes overlapping AA common variants should be removed as they act as confounders (as they may impact probe performance/binding/extension).

--line 168. Show the model and the program used to fit the model

--line 173. The cis window is very small. With this sample size, the analysis is well powered to examine larger cis windows, e.g. 500kb, 1Mb. Why was the small window selected? How does it impact all downstream analyses: heritability, colocalization, etc.

--line 184-191. The number of permutations seems too low to get an accurate distribution.

--line 192. Ideally, a backward-forward model (https://github.com/funpopgen/multiple_eqtl_mapping) needs to be used to estimate the independent signals jointly (see (GTEx Consortium 2020), applied to eQTLs)

--Line 304-305. Please describe the approach used to perform enrichment. E.g. did you consider the lead/primary variant for a mQTL CpG? What background was used? Why didn't you use an approach that a) take into account the meQTL signal landscape and b) models annotations jointly (e.g. TORUS: <https://github.com/xqwen/torus>. see (Xiong et al. 2021), a comparison between Fisher test and TORUS approach is performed)

--line 325. Typo: within the mQTL, I assume. Were the meQTLs recomputed for a 100kb window (since they were computed using a 50kb window)

--line 318-339. I appreciate the thorough scan for eQTL- and meQTL-tuned priors, reflecting the higher prior of finding a meQTL in the genome compared to eQTL.

Results

--line 411-428. It would be important to have a ballpark estimate of the fraction of meQTLs discovered in GENOA cohort that are not found in other meQTL maps, as well as the reason: what is the weight of the different factors into lack of replication? Is it mostly that GENOA is a well-powered cohort power compared to others, i.e. unreplicated meQTLs correspond to small meQTL effects, or unreplicated meQTLs correspond mostly to high MAF in AA and low MAF variants in other populations? The current cross-cohort comparison only shows that the mQTL findings in other cohorts

--line 439-444. Is that expected? Describe the findings in the context of what is known in the field and add pertinent references

--line 518-539. The molecular links to GWAS traits that involve meQTLs should be explored more thoroughly if possible (more traits), and convey the importance of ancestry-matching analysis.

--The section "Co-localization of eQTLs and meQTLs" involves colocalizations between eQTLs, mQTLs and GWAS.

--Only the colocalization involving the three traits (GWAS-eQTL-mQTL) are reported. There is benefit in identifying GWAS-meQTL links not supported by eQTLs, as they provide molecular support to the GWAS signal and, based on the location of the CpG, can point to a putative gene candidate. While the pairwise colocalization has been conducted (line 836) has not been reported in the Methods or results.

--The reported number of GWAS-eQTL-mQTL colocalizations is surprisingly small. Can the authors comment?

--The authors select GWASs mapped in AA populations, but they do not show the benefit of performing GWAS-QTL ancestry-matching colocalizations. It would be very interesting, and key to underscore the value and uniqueness of the ancestry-aware meQTL map they generated, to consider a set of traits for which GWASs mapped in AA and European/ancestry populations exist, and also include an (EPIC-array based preferably) blood meQTL map in Europeans, to evaluate the benefits of employing AA meQTLs, e.g. identifying AA meQTL-GWAS colocalizations missed in European-ancestry meQTL-GWAS colocalizations.

--line 581-586: The authors do not convey the limitations of the array-based CpG site selection bias: they constitute a very small fraction of the CpGs in the genome and they are selected mostly by overlap with gene regulatory regions. Any comparison with eQTL patterns need to be contextualized based on that fact, at least the limitation clearly stated. The fraction of genes with eQTL cannot be readily compared to the fraction of CpGs with mQTL as the genes have been profiled genome-wide, in contrast with the CpG sites. Same for the heritability estimates.

--line 598-613: The largest meQTL study in blood, GoDMC, not included. Relevant to evaluate the uniqueness of the AA meQTL map generated. While all the claims are accurate, as mentioned, the work does not convey the value of the AA meQTL map resource. What fraction of AA meQTLs identified are not present in the other studies? What is the reason? The largest meQTL resource in blood, GoDMC, is not utilized. Please add it in your comparisons.

--“The identified meCpG sites are enriched with hypermethylated CpG sites (OR=1.44, p value<2.23e-308) and are depleted with hypomethylated CpG sites (OR=0.7, p value<2.23e-308).” Hyper and hypo need to be clearly defined.

--“The conditional meQTLs reside further away from the CpG site compared to the primary meQTLs, though they are still enriched around the CpG site when compared with non-meQTLs”. the term “non-meQTL” is not intuitive here. please explain?

--This paper does not explore the number of CpGs linked to a given meQTL causal variant. and the possibility that a given causal mQTL variant can increase methylation at some CpGs while decreasing methylation at other nearby CpG sites.

--Fig1 needs an overall title

--I don't understand this statement, so may need some clarification: "The number of independent meQTLs across CpGs in the conditional analysis is positively correlated with the number of meQTLs in the unconditional analysis (Pearson's correlation coefficient =0.347, p-value<2.23e-308).

--based on the p-value (0.58) you cannot claim enrichment in shelves.

--what total number (and %) of meQTLs (and mCpGs) could potentially be attributable to (1) SNP disruption of CpG site and (2) SNP disruption of TF?

--no functional enrichment analyses based on SNP locations?

--"The CpG sites in the colocalized pairs where the common SNP displays opposite effects on expression and methylation are enriched in promoter regions (OR=1.48, p-value=6.57e-12), more so than those with the same direction of effects (OR=1.22, p-value=2.12e-03)." Unclear what the comparison group is here.

--"We carefully examined the eGene-meCpG pairs that are only significant in one model and thus have mediation effects with clear directionality." Not sure this is the case. Given that model may barely pass the threshold and the other model may be just below the threshold. Thus evidence for each model would be roughly similar.

Discussion/Conclusions

>5 million meQTLs. These are independent meQTLs? This implies that over half of the SNPs tested have a causal impact on DNA methylation. Unrealistic?

--"The colocalization and mediation analysis from the present study also support an important mediation role of cis-meQTL associated CpG sites in explaining and mediating gene expression levels in African Americans." Determining whether DNA methylation is a mediator or responsive to transcription factor activity is not possible using this type of data.

--“Comparing to the eQTL mapping study in the same cohort [9], we found that the proportion of CpG sites identified to harbor meQTLs (45.45%) is higher than the proportion of genes identified to harbor eQTLs (31.08%).” The sample size is same? Not sure this is a fair comparison given the differences in technologies used for expression vs. methylation measurement. Same point for the heritability comparisons that follow.

--“Comparing to the previous meQTL studies, we found a higher proportion of overlapped significant SNP-CpG pairs in African ancestry than other ancestries” Seems like many issues will complicate this comparison. Sample size and power of prior studies. Significance threshold used in prior studies.

Reviewer #2 (Remarks to the Author):

Identifying meQTLs will be important to understand genetic architecture for DNA methylation and help to understand clinical phenotypic variability. Many meQTL studies have been published. However, most of those studies mainly focused on samples from European ancestry cohorts. This study is the largest study using African ancestry cohorts. The meQTL resource provided by this study is valuable. I feel that maybe authors could comprehensively compare the results with European and other cohorts (such as Asian?).

Major

1. The two recently published meQTLs in large sample size have not been mentioned in this study.

Min, Josine L., et al. "Genomic and phenotypic insights from an atlas of genetic effects on DNA methylation." *Nature genetics* 53.9 (2021): 1311-1321.

Hawe, Johann S., et al. "Genetic variation influencing DNA methylation provides insights into molecular mechanisms regulating genomic function." *Nature genetics* 54.1 (2022): 18-29.

2. This paper is largely focused on cis-meQTLs. Is there any specific reason why did not include the trans-meQTLs?

Minor

1. In the abstract, it shows “5,004,406 cis-acting meQTLs in 359,306 meCpGs”. In the result, line 411, it shows “In total, we identified 359,306 meCpGs and 5,004,406 meQTLs”. I think it missed cis- in line 411. So it confused me that if the trans-meQTLs was reported or not?

2. Lines 416-422 are confusing. For example, “Specifically, compared to the FHS study of European ancestry [16], 2,440,659 (47.7%) significant SNP-CpG pairs detected there and tested in GENOA are also

significant in GENOA.” Does it mean that all the SNPs and CpGs overlapped in the two study, their SNP-CpG associations can be replicated? If it is case. Should I consider the replication ratio is 100%?

3. The conditional analysis part took lots of work. It is very valuable analysis to identify independent SNPs. That is an advantage in this study.

4. Usually, we would exclude CpGs including SNPs at a certain MAF at probes to avoid such artifacts. Line 490-497 made me confused if those CpGs were included in the results or not. Please clarify this.

REVIEWER COMMENTS

Reviewer #1 (Remarks to the Author):

1. This study reports the mapping of DNA methylation QTLs (mQTLs) in blood cells for a sample of 961 African American individuals. Analyses and approaches are sound. The primary novelty of this work is the focus of individuals of African ancestry, a group underrepresented in prior studies. The paper would benefit from an increased focus on the knowledge gained by specifically from a focus on AA for QTL mapping (e.g., for QTL discovery, coloc, mechanisms). The authors should consider more clearly defining which scientific insights are novel, with clearer comparisons with results from prior studies. The results are a bit too descriptive, and they needs more context and biological interpretation. Specific comments are below;

Thank you for your constructive comments. Following your comments, we have clearly defined the novel scientific findings, included comparison results with additional prior studies, and added more context and biological interpretations. Our detailed responses are listed below.

Abstract

2. --line 54-56. This finding is well-known, so should be noted here as consistent with prior studies.

We have added “, which is consistent with prior studies” to the Abstract (lines 57-58 on page 2).

3. --line 60. overlap or colocalize? Pairwise colocalization analysis goes beyond overlap as it aims to detect whether a causal variant is shared.

Indeed, it is “colocalize” instead of “overlap”. We have modified this sentence to read “Through co-localization analysis, we found that in our tested eGene-meCpG pairs, a large percentage of the cis-meQTLs colocalize with the expression quantitative-trait loci (cis-eQTL) from a previous study in the same population.” (lines 60-62 on page 2).

4. --line 63. Confusing sentence: what is the relation between mediation analysis and heritability?

We apologize for the confusion. We have reworded the sentence to make it clear that the heritability analysis and the mediation analysis are two separate analyses, which use cis-meQTLs and cis-meQTL associated CpG sites as exposures, respectively. The update sentence now reads: “Importantly, the identified cis-meQTLs explain a substantial proportion (median=24.6%) of methylation variation. In addition, in the mediation analysis, the cis-meQTL associated CpG sites mediate a substantial proportion (median=24.9%) of SNP effects underlying gene expression.” (lines 62-64 on page 2).

5. --line 64. SNP or variant (including small indels)?

We indeed removed insertions and deletions and retained only single nucleotide polymorphisms (SNP) for analyses. We have included this important information in the updated Methods section (line 147 on page 5).

Introduction:

6. --the authors appear to assume that DNA methylation mediates the effect of SNPs on gene expression. It is also possible that the epigenome responds to changes in gene expression/activity.

7. --the authors should acknowledge that DNA methylation represents chromatin conformation/structure and related histone modifications.

Thank you for these two comments. We have now included both of these important pieces of information in the Introduction, and we have cited relevant references: “Although variants in noncoding regions cannot directly influence the function of a gene by disrupting protein-coding sequence [1-3], they can influence gene expression through epigenetic regulatory mechanisms or induce epigenetic changes through the regulation of gene expression [4-6]. One important epigenetic change is DNA methylation, which influences gene expression by altering transcription factor binding capability, inducing conformational changes in the chromatin via histone modifications, and regulating microRNA expression [7, 8].” (lines 75 to 78 on page 3).

Methods

8. --line 161. The number of probes that passed QC seems on the high end. Authors should consider removing probes overlapping AA common variants should be removed as they act as confounders (as they may impact probe performance/binding/extension).

We have updated our QC steps following your suggestion. Specifically, in the previous version, we removed CpG sites on the X and Y chromosomes and then focused on a set of 790,636 CpG sites that contain at least one cis-SNP. In the updated manuscript, we now removed CpG sites on the X and Y chromosomes and followed [9] to further remove 36,860 cross-reactive probes [10] and 24,495 probes with a SNP at the target CpG site or within a single-base extension, resulting in a set of 771,134 CpG sites. This number is now very close to ref [9]. Afterwards, we further focused on 728,578 CpGs that contain at least one cis-SNP for meQTL mapping. These new QC details are provided in the updated Methods section (lines 164 to 167 on page 6) and all results are updated accordingly.

9. --line 168. Show the model and the program used to fit the model

We have added the statistical model and the command lines we used to fit the model in the updated Method section (lines 186-200 on page 6-7).

10. --line 173. The cis window is very small. With this sample size, the analysis is well powered

to examine larger cis windows, e.g. 500kb, 1Mb. Why was the small window selected? How does it impact all downstream analyses: heritability, colocalization, etc.

Thank you for the comment. We chose the cis window as +/-50kb of a CpG site for two reasons.

1. As is evident in previous studies [6, 11-16] as well as in our analysis (e.g. Figure 1), the detected cis-meQTLs are highly enriched around the CpG site. The strong enrichment pattern of cis-meQTLs near the CpG site is in direct contrast to the observations in the parallel eQTL studies, where the identified cis-eQTLs tend to reside in a much larger cis window centering around the transcription starting site (TSS). Therefore, unlike eQTL mapping studies where the window size is usually chosen to be 500kb or 1Mb, it is not necessary to move beyond 50kb of a CpG site for cis-meQTL mapping studies. Indeed, previous cis-meQTL studies also used short distances and only examined SNPs that reside within 3kb[11], 5kb[6], 20kb[17], or 50kb[18] of a CpG site.

To thoroughly investigate the impact of difference in the window size for cis-meQTL mapping results, we carried out additional analysis on Chr 22. (Due to the computational reason explained in point #2 below, we did not carry out the new analysis across the genome). Specifically, we varied the window size to be either +/-50kb, +/-500kb, or +/-1Mb on Chr 22 and carried out cis-meQTL mapping analysis. We found that the number of cis-meCpGs identified is 8740, 8150, and 7860 for the three window sizes, respectively. The cis-meCpGs share high overlap across the window sizes, with Jaccard index ranging from 0.79 to 0.93 (Figure S5). The number of detected cis-meCpGs slightly reduces with increasing window size as enlarging the window size increases the number of null signals and thus diluted the power to detect the true signals. In addition, the number of meQTLs is 76,884, 83,795, and 82,875 for the three different window sizes, respectively (Figure S5). The identified cis-meQTLs also share high overlap across the window sizes, with Jaccard index ranging from 0.82 to 0.96. The identified meQTLs only increased by 7.8% from +/-50kb to +/-1Mb (Figure S6).

We further examined the influence of different window sizes on heritability estimation (Figure S7) and found that the heritability estimates do not change much across different window sizes either (Figure S8-S9 and Table S2), suggesting that the most cis-meCpGs do reside within 50kb window. The details are updated in the Methods section in lines 176-180 on page 6 and the Results section in lines 506 to 507 on page 15.

2. Moving beyond 50kb can incur extremely heavy computation burden, due to the very large number of SNP-CpG pairs to be examined. Let's take a look at Chr 22 as an example. There are 16,349 CpG sites on Chr 22 (versus 728,578 CpG sites on the entire genome). When we use a cis-window of 50kb, we need to examine a total of 5,329,514 SNP-CpG pairs. When we use a cis-window of 1 Mb, the number of examined SNP-CpG pairs becomes 104,209,658, representing a 19-fold increase in the required computing resource. Indeed, it took us 20 hours to analyze a cis-window of 50kb on Chr 22 while took us 384 hours to analyze a cis-window of 1MB on Chr 22 on a single CPU thread.

Because of the above two reasons, we performed our main meQTL mapping analysis using a window size of 50kb.

11. --line 184-191. The number of permutations seems too low to get an accurate distribution.

Thank you for the comment. We primarily followed previous eQTL mapping studies [19] and used 10 permutations to construct the empirical null distribution of p-values. Previous eQTL mapping studies have used a variety of permutation numbers such as 10 [19, 20], 500 [21], or 1000 [22, 23] permutations. Previous meQTL mapping studies also used a variety of permutation numbers, such as 10 [24-26], 100 [27], or 1000 [28].

To thoroughly examine the influence of the number of permutations on meQTL mapping results, we performed additional mapping analysis on Chr 22. Specifically, we varied the number of permutations to be 2, 5, 10, 50, 100, 200, 300, 400, or 500. Regardless of the number of permutations, we found that the number of detected meCpGs is almost identical (mean=8,727, sd=25.7), with high overlap across the number of permutations (Jaccard index range: 0.997-0.9994, Figure S10). For example, we detected 8,740 meCpGs with 10 permutations and 8,774 meCpGs with 500 permutations, with almost identical results between the two (Jaccard index = 0.9972).

Importantly, running a large number of permutations incurs heavy computational burden. Specifically, it could take 1848 hours (77 days) to run 500 permutations using a single thread for the 16,349 CpG sites on Chr 22. It can become prohibitive to analyze 728,578 CpGs across the genome.

Because of the above reasons, we decided to use 10 permutations as our main analysis. We have also included the new results on Chr 22 and added additional sentences to explain the reasons for choosing 10 permutations in the updated Methods (lines 204 to 207 on page 7) and Results section (lines 507-509 on page 15).

12. --line 192. Ideally, a backward-forward model (https://github.com/funpopgen/multiple_eqtl_mapping) needs to be used to estimate the independent signals jointly (see (GTEx Consortium 2020), applied to eQTLs)

Previously we followed [19, 29] and used a forward model to detect independent meQTL signals in the conditional analysis. Following your comment, we examined the GTEx Consortium 2020 paper for eQTL mapping [30] and added the backward step to our forward model, and we have now tested each variant separately controlling for all other discovered variants in a CpG site and dropped the variant if no variant is significant in this CpG site.

Previously we identified a total of 700,754 independent meQTLs through conditional analysis using the forward model. The detected meQTLs include 341,448 conditional meQTLs and 359,306

primary meQTLs. Most meCpGs (54.3%) contain only one independent meQTL (i.e. primary meQTL) with some containing up to 26 independent meQTLs. After the new QC procedure (details in the response to your comment #4) and the added backward step, we now identified a total of 614,195 independent meQTLs in the conditional analysis, including 293,230 conditional meQTLs and the same 320,965 primary meQTLs. Most meCpGs (55%) contain only one independent meQTL with some containing up to 19 independent meQTLs. The details are updated in the Methods (lines 213 to 223 on page 7) and Results sections (line 572 on page 16).

13. --Line 304-305. Please describe the approach used to perform enrichment. E.g. did you consider the lead/primary variant for a mQTL CpG? What background was used? Why didn't you use an approach that a) take into account the meQTL signal landscape and b) models annotations jointly (e.g. TORUS: <https://github.com/xqwen/torus>. see (Xiong et al. 2021), a comparison between Fisher test and TORUS approach is performed)

Thank you for the comment. In the enrichment analysis, we considered all meQTLs in the analysis, which include all SNPs that passed the FDR threshold of 0.05. Therefore, the variants considered for enrichment analysis include both lead/primary variants and conditional variants. For background, we used all tested variants as background.

Following your comments, we performed a new analysis using Torus to take into account both the meQTL signal landscape and multiple annotations. We compared the results from Torus and Fisher's exact test and found them to be largely similar. The similarity in results between the two tests are in line with previous observations [31] (please note that we did not find the "Xiong et al. 2021" paper, but found this closely related ref). For example, in both tests, we found the meQTLs are enriched for active chromatin states, such as active transcription start site (TSS)-proximal promoter states (TssA, TssAFlnk), and enhancer states (Enh, EnhG). The only difference between the two tests is that Fisher's exact test identified ZNF/Rpts (zinc finger protein genes) and Quies (quiescent/low) to be slightly depleted in the meQTLs (which is consistent with ref [16]) while Torus identified them to be slightly enriched with meQTLs (Figure S19). The details are updated in the Methods (lines 374 to 376 on page 11) and Results sections (lines 606 to 609 on page 17).

14. --line 325. Typo: within the mQTL, I assume. Were the meQTLs recomputed for a 100kb window (since they were computed using a 50kb window)

We have corrected the typo. We also apologize for the miscommunication on the window size. We actually mapped SNPs that are +/-50kb to a CpG site, so the window size is 100kb. To make it clearer, we modified the sentence in line 177 on page 6 to be "we followed [32] to extract cis-SNPs that are within +/- 50kb of the CpG site".

15. --line 318-339. I appreciate the thorough scan for eQTL- and meQTL-tuned priors, reflecting the higher prior of finding a meQTL in the genome compared to eQTL.

Thank you!

Results

16. --line 411-428. It would be important to have a ballpark estimate of the fraction of meQTLs discovered in GENOA cohort that are not found in other meQTL maps, as well as the reason: what is the weight of the different factors into lack of replication? Is it mostly that GENOA is a well-powered cohort power compared to others, i.e. unreplicated meQTLs correspond to small meQTL effects, or unreplicated meQTLs correspond mostly to high MAF in AA and low MAF variants in other populations? The current cross-cohort comparison only shows that the mQTL findings in other cohorts

Thank you for the comments. Directly answering your question is unfortunately very challenging to do because these previous meQTL studies analyzed a different set of SNP-CpG pairs and reported only the significant SNP-CpG pairs but not all the tested pairs. Consequently, when some meQTLs identified in the present study were not “replicated” in the previous studies, we often do not know whether the previous studies did not identify these meQTLs or simply did not test these SNP-CpG pairs. However, we performed additional comparisons to indirectly address your question. Specifically, in each comparison, we calculated a new statistic, the replication rate (Storey’s π_1), which is defined as the expected true positive rate and is estimated by selecting the significant SNP-CpG pairs in each comparison cohort and examining their p value distribution in GENOA [23, 33]. Therefore, the replication rate effectively captures the proportion of signals in the previous study that is replicated in the present study. Again, we were unable to calculate the replication rate in the reverse way through selecting the significant SNP-CpG pairs in GENOA and examining their p value distribution in each comparison cohort, because the previous meQTL studies analyzed a different set of SNP-CpG pairs and reported only the significant SNP-CpG pairs but not all the tested pairs.

In the new analysis, we found that the replication rate in GENOA ranges from 0.9 to 0.93 at different cutoffs when compared to the Hawe et al. study with European and South Asian ancestries [28]. The meQTLs detected in either European or South Asian ancestry of the Hawe et al. study but not replicated in GENOA tend to have lower absolute effect sizes (median = 0.208 in comparison with EU and 0.198 with SA) than those detected in both studies (median=0.558 in comparison with EU and 0.538 with SA, Figure S12). The meQTLs detected in European or South Asian but not in GENOA also tend to have a lower allele frequency (median = 0.166 in comparison with EU, 0.17 with SA) to those detected in both studies (median=0.244 in comparison with EU, 0.247 with SA, Figure S12). In addition, the replication rate of GENOA ranges from 0.9 to 0.93 at different significant p value cutoffs when compared to the BEST study of South Asian ancestry [9]. The meQTLs detected in BEST but not in GENOA tend to have lower absolute effect sizes (median = 0.214) in GENOA than those detected in both studies (median=0.633, Figure S13). The meQTLs detected in South Asian individuals in BEST but not in GENOA tend to have a lower allele frequency (median = 0.223) to those detected in both studies (median=0.274, Figure S13). The replication rate of GENOA ranges from 0.76 to 0.77 at different significant p value cutoffs when

compared to the GoDMC study of European ancestry [16] (Table S3). The meQTLs detected in GoDMC but not in GENOA tend to have lower absolute effect sizes (median = 0.149) in GENOA than those detected in both studies (median=0.541, Figure S14). The allele frequencies in European individuals for the SNPs in SNP-CpG pairs detected in GoDMC but not in GENOA tend to have a lower allele frequency (median = 0.171) to those detected in both studies (median=0.245, Figure S14). The new analysis and results are updated in the Methods (line 238, lines 245 to 265 on page 8, and lines 304 to 327 on pages 9) and Results sections (lines 523 to 544 on page 15-16).

17. --line 439-444. Is that expected? Describe the findings in the context of what is known in the field and add pertinent references.

Yes, it is expected that the majority of PVE in methylation level is explained by trans-SNPs, with only a fraction explained by cis-SNPs. We have included a reference supporting this [34]. In [34], the average SNP-based heritability underlying methylation ranges between 0.20 and 0.25 over the life course, with the majority of the methylation variance explained by SNPs located over 1 Mb away from the CpG site. We have updated this sentence in the Results section to read “Consistent with [34, 35], we found that the majority of PVE in methylation level is explained by trans-SNPs, with only a fraction explained by cis-SNPs.” (lines 562 to 564 on page 16).

18. --line 518-539. The molecular links to GWAS traits that involve meQTLs should be explored more thoroughly if possible (more traits), and convey the importance of ancestry-matching analysis.

Thank you for your comment. We selected six GWAS traits for colocalization analysis because these were the only traits that we could find that (1) were conducted in the African ancestry and (2) made GWAS summary statistics across the entire genome publicly available. The six traits include type 2 diabetes (T2D) statistics from the MEta-analysis of type 2 Diabetes in African Americans (MEDIA) Consortium study, body mass index (BMI) from the African Ancestry Anthropometry Genetics Consortium (AAAGC) [36], and four blood pressure traits including pulse pressure (PP), diastolic blood pressure (DBP), systolic blood pressure (SBP), and hypertension (HTN) statistics from COGENT-BP meta-analyses [37]. Therefore, we were unable to obtain additional traits for analysis to address the first point of your comment.

For the second point of your comment on ancestry matching analysis, we performed additional analysis using a matched number of European samples from the UK Biobank (UKBB) for these traits. Specifically, we focused on three out of the six traits that include SBP, DBP, and BMI, as we did not have access to HTN, PP, and the phecode of T2D in our UKBB application. For each of the three traits in turn, we obtained a random subsample of individuals of white British ancestry to match the sample size of the African ancestry GWAS studies (i.e. 53K for BMI, and 32K for SBP and DBP). We then performed GWAS analysis where we included age, sex, and top 20 genetic PCs as covariates to obtain summary statistics. Afterwards, we performed colocalization analysis with coloc (between GWAS and meQTLs, and between GWAS and eQTLs) and performed multi-

trait colocalization analysis with moloc (among GWAS, meQTLs and eQTLs), using the GWAS summary statistics from UKBB. In the analysis, we identified three multi-trait colocalizations with the African ancestry GWAS (1 for SBP, 2 for DBP) and one with the European ancestry GWAS (for BMI). We also identified three GWAS-meQTL colocalization signals (1 for BMI and 2 for DBP) with the African Ancestry GWAS, and five with the European Ancestry GWAS (2 for BMI and 3 for DBP). However, the number of identified signals was quite small, suggesting that much larger GWAS of African ancestry are needed in the future to arrive at any definitive conclusions. These new analyses are updated in the Methods (lines 426 to 434 on page 12) and Results sections (lines 667 to 679 on page 19).

19. --The section “Co-localization of eQTLs and meQTLs” involves colocalizations between eQTLs, mQTLs and GWAS.

Thank you for spotting the error. We have updated the section name to be “Co-localization of eQTLs, meQTLs, and GWAS variants.” (line 627 on page 18).

20. --Only the colocalization involving the three traits (GWAS-eQTL-mQTL) are reported. There is benefit in identifying GWAS-meQTL links not supported by eQTLs, as they provide molecular support to the GWAS signal and, based on the location of the CpG, can point to a putative gene candidate. While the pairwise colocalization has been conducted (line 836) has not been reported in the Methods or results.

We previously performed pairwise colocalization analysis to identify colocalization signals for GWAS-meQTL using the meQTLs used in the GWAS-eQTL-meQTL colocalization analysis. The results were included in the previous Supplementary Table S5. Briefly, we identified 4 meQTL-GWAS colocalization signals using coloc – 1 of them is also identified in eQTL-GWAS and another 1 is also identified in GWAS-eQTL-meQTL analysis. We also identified 6 meQTL-GWAS colocalization signals using Susie – 5 of them are also identified in eQTL-GWAS colocalization while none of them are identified in GWAS-eQTL-meQTL analysis. Following your comments, we further expanded our analysis to using genome-wide meQTLs. The new analysis identified 0, 8, 1, 4, 1, and 0 colocalized GWAS-meQTL signals for SBP, BMI, DBP, HTN, PP, and T2D, respectively. However, none of these co-localized meQTL were associated with gene expression levels. We have included the new analysis and results to the updated Methods (lines 411 to 424 on page 12) and Results sections (lines 667 to 679 on page 19).

21. --The reported number of GWAS-eQTL-mQTL colocalizations is surprisingly small. Can the authors comment?

We reviewed the literature and found that previous studies also identified a similarly small number of GWAS-eQTL-meQTL colocalizations using moloc. Specifically, we identified 1 GWAS-eQTL-meQTL colocalization for SBP, 2 for DBP, and 1 for PP. While previous studies identified 5 GWAS-eQTL-meQTL colocalizations for FEV1 [38], 2 for neuroticism, 4 for schizophrenia, 4 for

education, 1 for Alzheimer's disease, 4 for insomnia [39], 5 for eGFR [40], 0 for adult onset asthma, 22 for childhood onset asthma, and 2 for asthma with onset at all ages [41]. Therefore, the reported numbers in the present study are largely consistent with previous studies.

We initially thought that one potential explanation behind the relatively low number of colocalization signals from moloc is that moloc makes a restrictive assumption that there is a single causal variant underlying all associations. However, we observed similarly small number of colocalization signals for the pairwise colocalization analysis using Susie, which relaxes the single causal variant assumption. Therefore, we simply stated the above results in the updated Results section (lines 650 to 651 on page 18) without further investigating the reason behind the small number of colocalization signals commonly observed in moloc analysis.

22. --The authors select GWASs mapped in AA populations, but they do not show the benefit of performing GWAS-QTL ancestry-matching colocalizations. It would be very interesting, and key to underscore the value and uniqueness of the ancestry-aware meQTL map they generated, to consider a set of traits for which GWASs mapped in AA and European/ancestry populations exist, and also include an (EPIC-array based preferably) blood meQTL map in Europeans, to evaluate the benefits of employing AA meQTLs, e.g. identifying AA meQTL-GWAS colocalizations missed in European-ancestry meQTL-GWAS colocalizations.

Thank you for your comment. We have included GWAS-meQTL ancestry-matching colocalizations with samples subsampled from UKBB. Please refer to our response to your comment #18 for details.

Please note that in our response to your comment #18, we performed ancestry-matching analysis using GWAS of European ancestry and GWAS of African ancestry. We have also tried to perform ancestry-matching analysis using meQTL mapping studies of European ancestry and meQTL mapping studies of African ancestry, but failed to do so due to a lack of such studies measured on similar methylation arrays with comparable sample sizes. For example, the study in [42] performed meQTL mapping with Illumina EPIC array in whole blood on samples from the UK Household Longitudinal study (N=1,111), and posted their data at <https://www.epigenomicslab.com/online-data-resources/>. However, we were unable to open the subpage entitled "Methylomic profiling: blood and brain" to access the supplementary summary statistics in the paper. The study in [27] performed meQTL mapping with Illumina EPIC array in monocytes on samples of European (N=78) and African (N=78) descent, but the sample size in European ancestry in the study is too small. The study in [43] measured methylation profile with Illumina 450k array in the whole blood on samples from the Northern Sweden Population Health Study (N=743). Although the sample size is somewhat comparable to ours, they aimed to identify "vSNPs" associated with variance heterogeneity in DNA methylation levels rather than traditional meQTLs. The study in [44] performed meQTL mapping with Illumina 450k array in peripheral blood on samples from Mexican American individuals (N=850). Though the sample size is also comparable, their study population is not from European ancestry, thus making it hard to perform ancestry matching. Therefore, we were unable to perform ancestry-matching analysis

using meQTL mapping studies of European ancestry and meQTL mapping studies of African ancestry.

23. --line 581-586: The authors do not convey the limitations of the array-based CpG site selection bias: they constitute a very small fraction of the CpGs in the genome and they are selected mostly by overlap with gene regulatory regions. Any comparison with eQTL patterns need to be contextualized based on that fact, at least the limitation clearly stated. The fraction of genes with eQTL cannot be readily compared to the fraction of CpGs with mQTL as the genes have been profiled genome-wide, in contrast with the CpG sites. Same for the heritability estimates.

Following your comment, we have added the following information in the comparison between eQTLs and meQTLs to clearly state the limitation of the array-based CpG site selection bias, along with another limitation of using different tissues, in the updated Discussion section (lines 740 to 751 on page 21):

“While the sample sizes of the eQTL study (N=1,032) and the meQTL study (N=961) are similar, we acknowledge that many other factors may influence the comparison of results and contribute to the observed differences. First, the tissues for the samples in the expression and methylation data are different: the expression data are collected from lymphoblastoid cell lines (LCLs), while the methylation data are collected from peripheral blood leukocytes. Second, the technologies used for the measurement are different. The expression were measured using the Affymetrix Human Transcriptome Array 2.0, which surveys the entire transcriptome. In contrast, the methylation was assessed using the Illumina Infinium HumanMethylationEPIC BeadChip, which covers only 30% of the human methylome with a particular focus on the regulatory regions [10]. The limited methylome coverage should be taken into consideration when generalizing the meQTL findings to the whole genome [45].”

24. --line 598-613: The largest meQTL study in blood, GoDMC, not included. Relevant to evaluate the uniqueness of the AA meQTL map generated. While all the claims are accurate, as mentioned, the work does not convey the value of the AA meQTL map resource. What fraction of AA meQTLs identified are not present in the other studies? What is the reason? The largest meQTL resource in blood, GoDMC, is not utilized. Please add it in your comparisons.

We have included the GoDMC study [16] in our comparison. Please refer to our reply to your comment #16 for details.

25. --“The identified meCpG sites are enriched with hypermethylated CpG sites (OR=1.44, p value<2.23e-308) and are depleted with hypomethylated CpG sites (OR=0.7, p value<2.23e-308).” Hyper and hypo need to be clearly defined.

We previously defined hyper- and hypo- briefly in the paragraph above this sentence. However, to make things more clear, we have added a phrase to the sentence: ", where hypermethylated and hypomethylated CpG sites are defined as CpG sites with a beta value above 0.7 or below 0.3, respectively [46]." (lines 511 to 512 on page 15).

26. --“The conditional meQTLs reside further away from the CpG site compared to the primary meQTLs, though they are still enriched around the CpG site when compared with non-meQTLs”. the term “non-meQTL” is not intuitive here. please explain?

The “non-meQTL” refers to a SNP that is not significant in any SNP-CpG pairs tested. That is, its smallest p-value across all the SNP-CpG pairs is above the p value threshold $2.267195e-4$, which corresponds to an empirical FDR of 5%. We have added this explanation (lines 580 to 581 on page 17).

27. --This paper does not explore the number of CpGs linked to a given meQTL causal variant. and the possibility that a given causal mQTL variant can increase methylation at some CpGs while decreasing methylation at other nearby CpG sites.

Thank you for the comment. We explored the number of CpGs linked to an meQTL and summarized the number of CpGs positively/negatively associated with the meQTL. Indeed, for a given meQTL, it can increase methylation at some CpGs while decreasing methylation at other CpGs. Among the 4,565,687 unique meQTLs, the number of CpGs associated with an meQTL ranges from 1 to 166 (median=2, mean=3.4). The number of CpGs positively associated with an meQTL ranges from 0 to 118 (median=1, mean=1.7) while the number of CpGs negatively associated with an meQTL ranges from 0 to 83 (median=1, mean=1.7). The median proportion of positively associated CpGs for an meQTL is 0.5 (mean=0.49, Figure S11). We have included these results in the updated Results section (lines 513 to 519 on page 15).

28. --Fig1 needs an overall title

We added the overall title for figure 1 as “Characterization of the identified meQTLs”.

29. --I don’t understand this statement, so may need some clarification: “The number of independent meQTLs across CpGs in the conditional analysis is positively correlated with the number of meQTLs in the unconditional analysis (Pearson’s correlation coefficient =0.347, p-value<2.23e-308).

We apologize for the confusion. We carried out two meQTL mapping analyses for each CpG site. The first analysis is the standard marginal/unconditional analysis, where we simply counted the number of meQTLs that pass the significance level regardless of whether they are correlated with each other or not. This is the number of meQTLs in the unconditional analysis in that sentence.

The second analysis is the conditional analysis, where we carried out forward-backward selection to identify independent meQTLs associated with the CpG site. This is the number of independent meQTLs in the conditional analysis. With the two analyses, we calculated the Pearson's coefficient between the number of all meQTLs with the number of independent meQTLs across all meCpG sites to obtain the Pearson's correlation coefficient. We have reworded the statement to read "The number of meQTLs in the standard and conditional analyses are positively correlated with each other (Pearson's correlation coefficient=0.347, p-value<2.23e-308)." (lines 581 to 583 on page 17).

30. --based on the p-value (0.58) you cannot claim enrichment in shelves.

Thank you for spotting the error. We have deleted this part of the conclusion in the sentence (previous line 471 on page 13).

31. --what total number (and %) of meQTLs (and mCpGs) could potentially be attributable to (1) SNP disruption of CpG site and (2) SNP disruption of TF?

Thank you for your comment. For your two questions:

(1) We previously tested 790,636 CpGs and 9,072,756 SNPs, among which 359,306 and 5,004,406 were identified as meCpGs and meQTLs, respectively. Among the tested CpG sites, 24,495 of them are directly disrupted by a SNP, among which 20,384 are meCpGs. Among the tested SNPs, 24,802 SNPs directly disrupt CpGs, among which 20,584 are meQTLs. Therefore, 5.67% of meCpGs and 0.4% of meQTLs could potentially be attributed to SNP disruption. In the updated manuscript, we have removed these SNPs that disrupt CpG sites to address your comment #8 and reviewer2's minor comment #4.

(2) Through the new QC procedure (details in our reply to your comment #8), we now detected 4,565,687 unique meQTLs, among which 43,115 (0.944%) disrupt TF binding motifs (note that a total of 79,331 SNPs disrupt TF binding motifs). Among the 609,992 independent meQTLs obtained from conditional analysis, 7,445 (1.22%) of them disrupt TF binding motifs. We have added these results to the updated Results section (lines 622 to 624 on page 18).

32. --no functional enrichment analyses based on SNP locations?

In the updated manuscript, we performed functional enrichment analysis for meQTLs based on summary statistics and SNP location information with Torus. The results are provided in the response to your comment #13. Briefly, we found the meQTLs are enriched for active chromatin states, such as active transcription start site (TSS)-proximal promoter states (TssA, TssAFlnk), and enhancer states (Enh, EnhG). The details are updated in the Methods (lines 374 to 376 on page 11) and Results sections (lines 606 to 609 on page 17).

33. --“The CpG sites in the colocalized pairs where the common SNP displays opposite effects on expression and methylation are enriched in promoter regions (OR=1.48, p-value=6.57e-12), more so than those with the same direction of effects (OR=1.22, p-value=2.12e-03).” Unclear what the comparison group is here.

We used CpGs in the tested eGene-meCpG pairs for colocalization as background. We have added this sentence in lines 641 to 642 on page 18.

34. --“We carefully examined the eGene-meCpG pairs that are only significant in one model and thus have mediation effects with clear directionality.” Not sure this is the case. Given that model may barely pass the threshold and the other model may be just below the threshold. Thus evidence for each model would be roughly similar.

Thank you for the comment. We have included additional details in the updated manuscript. Specifically, in the new analysis with the updated QC process, we identified two eGene-meCpG pairs that have an adjusted q value < 0.05 in the SME analysis but have an adjusted q value > 0.1 in the SEM analysis: CHKB-CPT1B/cg00047287 and TUBD1/cg02095219. In particular, the CpG site cg00047287 is located in the CpG island and TSS200 of the CPT1B gene, and is significant in mediating the effect of rs8137478 on CHKB-CPT1B (SME p-value= 0.0015, adjusted q value 0.03, Figure 6A), which is a potential target for age-dependent intramuscular lipid accumulation and insulin resistance [47]. The CpG cg02095219 is located in the open sea near the TUBD1 gene, and is significant in mediating the effect of rs111282327 on TUBD1 (SME p-value= 0.002, adjusted q value 0.04, Figure 6B), whose expression is associated with copy number status in primary breast tumors [48]. In addition, we also identified one eGene-meCpG pair that has an adjusted q value > 0.1 in the SME analysis but has an adjusted q value < 0.05 in the SEM analysis: MB21D2/cg08100000. In particular, the CpG site cg08100000 is located in the gene body of MB21D2 and is significant in mediating the effect of rs13069487 on cg08100000 (SEM p value=0.0093, adjusted p value 0.024, Figure 7). The MB21D2 gene was identified to be a differentially methylated region (DMR) for human squamous cell carcinoma [49], and overexpression of MB21D2 can facilitate cell proliferation and invasion in cancer [50]. These details are updated in the Results sections (lines 697 to 712 on pages 19-20). In addition, following the essence of your comment, we have also weakened the overall conclusion throughout the text to not emphasize on the directionality of the mediation effects.

Discussion/Conclusions

35. >5 million meQTLs. These are independent meQTLs? This implies that over half of the SNPs tested have a causal impact on DNA methylation. Unrealistic?

The 5,004,406 meQTLs we detected are not independent meQTLs. These are the meQTLs with a significant marginal effect, many of which are in high LD with each other. We reported these

meQTLs following the FHS meQTL study [51] and Hannon et al. [42]. Specifically, the FHS study tested 8.5 million SNPs, and identified 4.7 million cis-meQTL SNPs at the Bonferroni-corrected multiple-testing threshold ($P < 2e-11$). Hannon et al. [42] tested 5.2 million genetics variants and identified 2.9 million cis-meQTL variants at Bonferroni-corrected multiple-testing threshold ($6.52e-14$). We have included these two citations and made it clear that these were not independent meQTLs. In addition, we have also included the number of lead/primary meQTLs and the number of independent meQTLs following the above sentence. Lead/primary meQTLs are defined as the meQTLs with the smallest p value for each meCpG site, while the independent meQTLs were obtained through conditional analysis. The updated sentence is in lines 716-717 on page 21.

36. --“The colocalization and mediation analysis from the present study also support an important mediation role of cis-meQTL associated CpG sites in explaining and mediating gene expression levels in African Americans.” Determining whether DNA methylation is a mediator or responsive to transcription factor activity is not possible using this type of data.

We agree. We have changed that part of conclusion to be “The colocalization and mediation analyses from the present study also provide evidence supporting co-regulation of methylation and gene expression in African Americans.” (lines 718 to 720 on page 21, lines 65-66 on page 2 and lines 115-116 on page 4).

37. --“Comparing to the eQTL mapping study in the same cohort [9], we found that the proportion of CpG sites identified to harbor meQTLs (45.45%) is higher than the proportion of genes identified to harbor eQTLs (31.08%).” The sample size is same? Not sure this is a fair comparison given the differences in technologies used for expression vs. methylation measurement. Same point for the heritability comparisons that follow.

Thank you for the comment. Yes, the sample size is similar, though the comparison is indeed influenced by technology differences and other factors. Therefore, following this comment and your comment #23, we have added the following information in the Discussion to clearly state the limitations of such comparison (lines 740 to 751 on page 21):

“While the sample sizes of the eQTL study ($N=1,032$) and the meQTL study ($N=961$) are similar, we acknowledge that many other factors may influence the comparison of results and contribute to the observed differences. First, the tissues for the samples in the expression and methylation data are different: the expression data are collected from lymphoblastoid cell lines (LCLs), while the methylation data are collected from peripheral blood leukocytes. Second, the technologies used for the measurement are different. The expression were measured using the Affymetrix Human Transcriptome Array 2.0, which surveys the entire transcriptome. In contrast, the methylation was assessed using the Illumina Infinium HumanMethylationEPIC BeadChip, which covers only 30% of the human methylome with a particular focus on the regulatory regions [10]. The limited methylome coverage should be taken into consideration when generalizing the

meQTL findings to the whole genome [45]. Both differences may influence the comparison results.”

38. --“Comparing to the previous meQTL studies, we found a higher proportion of overlapped significant SNP-CpG pairs in African ancestry than other ancestries” Seems like many issues will complicate this comparison. Sample size and power of prior studies. Significance threshold used in prior studies.

We agree. Indeed, in the previous version of the manuscript, we listed factors that may complicate the comparison: “This may be in part because the PB and CBA samples have a more similar ancestral background and LD structure than the FHS and BEST samples when compared with GENOA samples. Also, the FHS study has a much larger sample size which allowed them to detect meQTLs with small effects that are not detectable by other studies. It also may be a direct result of the methylation platforms used, as the meQTL studies in African ancestry samples were conducted using the 27K BeadChip, which is comprised almost exclusively of methylation sites in gene promoters and CpG islands [52]. The FHS and BEST meQTL studies were conducted using the 450K array that interrogates a much broader array of CpG types across the epigenome. We also note that we did not perform tests on all of the significant SNP-CpG pairs obtained from other studies. This is because we selected different cis-window sizes for the SNP-CpG pairs and used different methylation platforms.” (lines 757 to 767 on pages 21-22).

Following your comment, we performed additional analysis to address the potential effect of the p value significance thresholds used in the prior studies. To do so, we explored using three different thresholds: 1. The p value threshold used in the original compared study; 2. The p value threshold of $1e-5$ for both studies; 3. Bonferroni corrected p value <0.05 for available SNP-CpG pairs tested in both studies. In the analysis, we found that the replication rates are very similar (Table S3). The details are updated in the Methods section in lines 322 to 325 on page 9, and the Results section in lines 521 to 544 on pages 15-16.

Reviewer #2 (Remarks to the Author):

Identifying meQTLs will be important to understand genetic architecture for DNA methylation and help to understand clinical phenotypic variability. Many meQTL studies have been published. However, most of those studies mainly focused on samples from European ancestry cohorts. This study is the largest study using African ancestry cohorts. The meQTL resource provided by this study is valuable. I feel that maybe authors could comprehensively compare the results with European and other cohorts (such as Asian?).

Thank you for your positive comments. Following your general comment, we have added a comprehensive comparison with the GoDMC study, which is the largest meQTL mapping in European ancestry (27,750 Europeans) [16], and with the Hawe et al. study, which is largest meQTL mapping study in South Asian ancestry (3799 Europeans and 3195 South Asians) [28]. Details are provided below.

Major

1. The two recently published meQTLs in large sample size have not been mentioned in this study.

Min, Josine L., et al. "Genomic and phenotypic insights from an atlas of genetic effects on DNA methylation." *Nature genetics* 53.9 (2021): 1311-1321.

Hawe, Johann S., et al. "Genetic variation influencing DNA methylation provides insights into molecular mechanisms regulating genomic function." *Nature genetics* 54.1 (2022): 18-29.

We have added these two references to the Introduction section in lines 103 to 106 on page 3. We also performed comprehensive comparison with these two studies.

Specifically, the replication rate in GENOA ranges from 0.9 to 0.93 at different cutoffs when compared to the Hawe et al. study with European and South Asian ancestries [28]. The meQTLs detected in either European or South Asian ancestry of the Hawe et al. study but not replicated in GENOA tend to have lower absolute effect sizes (median = 0.208 in comparison with EU and 0.198 with SA) than those detected in both studies (median=0.558 in comparison with EU and 0.538 with SA, Figure S12). The meQTLs detected in European or South Asian but not in GENOA also tend to have a lower allele frequency (median = 0.166 in comparison with EU, 0.17 with SA) to those detected in both studies (median=0.244 in comparison with EU, 0.247 with SA, Figure S12). In addition, the replication rate of GENOA ranges from 0.9 to 0.93 at different significant p value cutoffs when compared to the BEST study of South Asian ancestry [9]. The meQTLs detected in BEST but not in GENOA tend to have lower absolute effect sizes (median = 0.214) in GENOA than those detected in both studies (median=0.633, Figure S13). The meQTLs detected in South Asian individuals in BEST but not in GENOA tend to have a lower allele frequency (median = 0.223) to those detected in both studies (median=0.274, Figure S13). The replication rate of GENOA ranges from 0.76 to 0.77 at different significant p value cutoffs when compared to the GoDMC study of European ancestry [16] (Table S3). The meQTLs detected in GoDMC but not in GENOA tend to have lower absolute effect sizes (median = 0.149) in GENOA than those detected

in both studies (median=0.541, Figure S14). The allele frequencies in European individuals for the SNPs in SNP-CpG pairs detected in GoDMC but not in GENOA tend to have a lower allele frequency (median = 0.171) to those detected in both studies (median=0.245, Figure S14). The new analysis and results are updated in the Methods (line 238, lines 245 to 265 on page 8, and lines 304 to 327 on page 9) and Results sections (lines 521 to 544 on pages 15-16).

We also performed additional analysis to address the potential effect of different p value significance thresholds used in the prior studies. To do so, we explored using three different thresholds: 1. The p value threshold used in the original compared study; 2. The p value threshold of 1e-5 for both studies; 3. Bonferroni corrected p value <0.05 for available SNP-CpG pairs tested in both studies) and found that the replication rates are very close (Table S3). The details are updated in the Methods section in lines 322 to 325 on page 9, and the Results section in lines 521 to 544 on pages 15-16.

2. This paper is largely focused on cis-meQTLs. Is there any specific reason why did not include the trans-meQTLs?

We focused on mapping cis-meQTLs for the following three reasons.

1. Identifying trans-meQTLs can incur extremely heavy computation burden, due to the very large number of SNP-CpG pairs to be examined. To see this, we performed trans-meQTL mapping analysis on randomly selected 500 CpG sites on Chr 22. In the analysis, we defined trans-SNPs as SNPs that reside either >50kb (because we defined cis- as <50kb) or >1 Mb (references: [16, 28]) to a CpG site. For this analysis, it took us 6 hours on a single thread. Therefore, it would take us 506 years to finish the trans-meQTL mapping analysis for all CpG sites on the entire genome using a single thread.

2. Identifying trans-meQTLs also requires very large disk usage. Even though we only saved summary statistics from the meQTL mapping in compressed files, it still took 350Mb disk space for each CpG site in the trans-meQTL mapping analysis. Therefore, if we were to save all results from trans-meQTL mapping for all CpG sites across the entire genome, it would easily take up >30Tb disk space even without any additional downstream analyses.

3. We briefly explored the results from the trans-meQTL mapping analysis for the 500 CpGs on Chr 22 mentioned above. Consistent with previous studies [28], we found that the number of identified trans-meQTLs is relatively small (137 trans-meQTLs) after Bonferroni correction. In addition, all of the identified trans-meQTLs are from the same chromosome (Table S9).

Therefore, we were unable to carry out trans-meQTL analysis without substantial improvement of the computing infrastructure at our university. We have included the above reasoning and results to the updated Discussion section (lines 779 to 781 on page 22).

Minor

1. In the abstract, it shows “5,004,406 cis-acting meQTLs in 359,306 meCpGs”. In the result, line 411, it shows “In total, we identified 359,306 meCpGs and 5,004,406 meQTLs”. I think it missed cis- in line 411. So it confused me that if the trans-meQTLs was reported or not?

Thank you for your comment, we added the “cis-” in line 513.

2. Lines 416-422 are confusing. For example, “Specifically, compared to the FHS study of European ancestry [16], 2,440,659 (47.7%) significant SNP-CpG pairs detected there and tested in GENOA are also significant in GENOA.” Does it mean that all the SNPs and CpGs overlapped in the two study, their SNP-CpG associations can be replicated? If it is case. Should I consider the replication ratio is 100%?

There are 27,235,69 significant SNP-CpG pairs available in the FHS study, and 10,894,656 of them have SNPs within 50kb of the CpG site. Among these 10,894,656 pairs, 5,112,264 of them were also tested in GENOA, and 2,440,659 of them are significant in GENOA. Following your comments, we have updated the replication analysis and calculated the replication rate at various p value thresholds. The details are provided in the reply to your major comment #1.

3. The conditional analysis part took lots of work. It is very valuable analysis to identify independent SNPs. That is an advantage in this study.

Thank you!

4. Usually, we would exclude CpGs including SNPs at a certain MAF at probes to avoid such artifacts. Line 490-497 made me confused if those CpGs were included in the results or not. Please clarify this.

Thank you for your comment. We have updated our QC steps following your suggestions. Specifically, in the previous version, we removed CpG sites on the X and Y chromosomes and then focused on a set of 790,636 CpG sites that contain at least one cis-SNP. In the updated manuscript, we now removed CpG sites on the X and Y chromosomes and followed [9] to further remove 36,860 cross-reactive probes [10], 24,495 probes with SNP at target CpG site or within a single-base extension, resulting in a set of 771,134 CpG sites. This number is now very close to ref [9]. Afterwards, we further focused on 728,578 CpGs that contain at least one cis-SNP for meQTL mapping. These new QC details are provided in the updated Methods section (lines 164 to 167 on page 6) and all results are updated accordingly.

References:

1. Gallagher MD, Chen-Plotkin AS. The Post-GWAS Era: From Association to Function. *Am J Hum Genet.* 2018;102(5):717-30. doi: 10.1016/j.ajhg.2018.04.002. PubMed PMID: 29727686; PubMed Central PMCID: PMC5986732.
2. Hindorff LA, Sethupathy P, Junkins HA, Ramos EM, Mehta JP, Collins FS, et al. Potential etiologic and functional implications of genome-wide association loci for human diseases and traits. *Proc Natl Acad Sci U S A.* 2009;106(23):9362-7. Epub 2009/05/27. doi: 10.1073/pnas.0903103106. PubMed PMID: 19474294; PubMed Central PMCID: PMC2687147.
3. Nica AC, Montgomery SB, Dimas AS, Stranger BE, Beazley C, Barroso I, et al. Candidate Causal Regulatory Effects by Integration of Expression QTLs with Complex Trait Genetic Associations. *Plos Genetics.* 2010;6(4). doi: ARTN e1000895 10.1371/journal.pgen.1000895. PubMed PMID: WOS:000277354200012.
4. Jin B, Robertson KD. DNA methyltransferases, DNA damage repair, and cancer. *Adv Exp Med Biol.* 2013;754:3-29. Epub 2012/09/08. doi: 10.1007/978-1-4419-9967-2_1. PubMed PMID: 22956494; PubMed Central PMCID: PMC3707278.
5. Mostowska A, Sajdak S, Pawlik P, Lianeri M, Jagodzinski PP. DNMT1, DNMT3A and DNMT3B gene variants in relation to ovarian cancer risk in the Polish population. *Mol Biol Rep.* 2013;40(8):4893-9. Epub 2013/05/15. doi: 10.1007/s11033-013-2589-0. PubMed PMID: 23666104; PubMed Central PMCID: PMC3723978.
6. Gutierrez-Arcelus M, Lappalainen T, Montgomery SB, Buil A, Ongen H, Yurovsky A, et al. Passive and active DNA methylation and the interplay with genetic variation in gene regulation. *Elife.* 2013;2:e00523. Epub 2013/06/12. doi: 10.7554/eLife.00523. PubMed PMID: 23755361; PubMed Central PMCID: PMC3673336.
7. Moore LD, Le T, Fan G. DNA methylation and its basic function. *Neuropsychopharmacology.* 2013;38(1):23-38. Epub 2012/07/12. doi: 10.1038/npp.2012.112. PubMed PMID: 22781841; PubMed Central PMCID: PMC3521964.
8. Phillips T. The role of methylation in gene expression. *Nature Education.* 2008;1.1:116.
9. Pierce BL, Tong L, Argos M, Demanelis K, Jasmine F, Rakibuz-Zaman M, et al. Co-occurring expression and methylation QTLs allow detection of common causal variants and shared biological mechanisms. *Nat Commun.* 2018;9(1):804. Epub 2018/02/25. doi: 10.1038/s41467-018-03209-9. PubMed PMID: 29476079; PubMed Central PMCID: PMC5824840.
10. Pidsley R, Zotenko E, Peters TJ, Lawrence MG, Risbridger GP, Molloy P, et al. Critical evaluation of the Illumina MethylationEPIC BeadChip microarray for whole-genome DNA methylation profiling. *Genome Biol.* 2016;17(1):208. Epub 2016/10/09. doi: 10.1186/s13059-016-1066-1. PubMed PMID: 27717381; PubMed Central PMCID: PMC5055731.
11. Banovich NE, Lan X, McVicker G, van de Geijn B, Degner JF, Blischak JD, et al. Methylation QTLs are associated with coordinated changes in transcription factor binding, histone modifications, and gene expression levels. *PLoS Genet.* 2014;10(9):e1004663. Epub 2014/09/19. doi: 10.1371/journal.pgen.1004663. PubMed PMID: 25233095; PubMed Central PMCID: PMC4169251.

12. Plongthongkum N, van Eijk KR, de Jong S, Wang T, Sul JH, Boks MPM, et al. Characterization of Genome-Methylome Interactions in 22 Nuclear Pedigrees. *Plos One*. 2014;9(7). doi: ARTN e99313
10.1371/journal.pone.0099313. PubMed PMID: WOS:000339618600002.
13. Ng B, White CC, Klein HU, Sieberts SK, McCabe C, Patrick E, et al. An xQTL map integrates the genetic architecture of the human brain's transcriptome and epigenome. *Nat Neurosci*. 2017;20(10):1418-26. Epub 2017/09/05. doi: 10.1038/nn.4632. PubMed PMID: 28869584; PubMed Central PMCID: PMC5785926.
14. Do C, Lang CF, Lin J, Darbary H, Krupska I, Gaba A, et al. Mechanisms and Disease Associations of Haplotype-Dependent Allele-Specific DNA Methylation. *Am J Hum Genet*. 2016;98(5):934-55. Epub 2016/05/08. doi: 10.1016/j.ajhg.2016.03.027. PubMed PMID: 27153397; PubMed Central PMCID: PMC4863666.
15. Hannon E, Spiers H, Viana J, Pidsley R, Burrage J, Murphy TM, et al. Methylation QTLs in the developing brain and their enrichment in schizophrenia risk loci. *Nature Neuroscience*. 2016;19(1):48-+. doi: 10.1038/nn.4182. PubMed PMID: WOS:000367254400012.
16. Min JL, Hemani G, Hannon E, Dekkers KF, Castillo-Fernandez J, Luijk R, et al. Genomic and phenotypic insights from an atlas of genetic effects on DNA methylation. *Nat Genet*. 2021;53(9):1311-21. Epub 2021/09/09. doi: 10.1038/s41588-021-00923-x. PubMed PMID: 34493871; PubMed Central PMCID: PMC7612069.
17. Lin DD, Chen JY, Perrone-Bizzozero N, Bustillo JR, Du YH, Calhoun VD, et al. Characterization of cross-tissue genetic-epigenetic effects and their patterns in schizophrenia. *Genome Medicine*. 2018;10. doi: ARTN 13
10.1186/s13073-018-0519-4. PubMed PMID: WOS:000426388400001.
18. Smith AK, Kilaru V, Kocak M, Almli LM, Mercer KB, Ressler KJ, et al. Methylation quantitative trait loci (meQTLs) are consistently detected across ancestry, developmental stage, and tissue type. *BMC Genomics*. 2014;15:145. Epub 2014/02/22. doi: 10.1186/1471-2164-15-145. PubMed PMID: 24555763; PubMed Central PMCID: PMC4028873.
19. Shang L, Smith JA, Zhao W, Kho M, Turner ST, Mosley TH, et al. Genetic Architecture of Gene Expression in European and African Americans: An eQTL Mapping Study in GENOA. *Am J Hum Genet*. 2020;106(4):496-512. Epub 2020/03/30. doi: 10.1016/j.ajhg.2020.03.002. PubMed PMID: 32220292; PubMed Central PMCID: PMC7118581.
20. van der Wijst MGP, Brugge H, de Vries DH, Deelen P, Swertz MA, LifeLines Cohort S, et al. Single-cell RNA sequencing identifies celltype-specific cis-eQTLs and co-expression QTLs. *Nat Genet*. 2018;50(4):493-7. Epub 2018/04/04. doi: 10.1038/s41588-018-0089-9. PubMed PMID: 29610479; PubMed Central PMCID: PMC5905669.
21. Moreno V, Alonso MH, Closa A, Valles X, Diez-Villanueva A, Valle L, et al. Colon-specific eQTL analysis to inform on functional SNPs. *Br J Cancer*. 2018;119(8):971-7. Epub 2018/10/05. doi: 10.1038/s41416-018-0018-9. PubMed PMID: 30283144; PubMed Central PMCID: PMC6203735.
22. Consortium GT. Human genomics. The Genotype-Tissue Expression (GTEx) pilot analysis: multitissue gene regulation in humans. *Science*. 2015;348(6235):648-60. Epub 2015/05/09. doi: 10.1126/science.1262110. PubMed PMID: 25954001; PubMed Central PMCID: PMC4547484.

23. Consortium GT, Laboratory DA, Coordinating Center -Analysis Working G, Statistical Methods groups-Analysis Working G, Enhancing Gg, Fund NIHC, et al. Genetic effects on gene expression across human tissues. *Nature*. 2017;550(7675):204-13. Epub 2017/10/13. doi: 10.1038/nature24277. PubMed PMID: 29022597; PubMed Central PMCID: PMC5776756.
24. van Dongen J, Ehli EA, Jansen R, van Beijsterveldt CEM, Willemsen G, Hottenga JJ, et al. Genome-wide analysis of DNA methylation in buccal cells: a study of monozygotic twins and mQTLs. *Epigenetics Chromatin*. 2018;11(1):54. Epub 2018/09/27. doi: 10.1186/s13072-018-0225-x. PubMed PMID: 30253792; PubMed Central PMCID: PMC6156977.
25. Morrow JD, Glass K, Cho MH, Hersh CP, Pinto-Plata V, Celli B, et al. Human Lung DNA Methylation Quantitative Trait Loci Colocalize with Chronic Obstructive Pulmonary Disease Genome-Wide Association Loci. *Am J Respir Crit Care Med*. 2018;197(10):1275-84. Epub 2018/01/10. doi: 10.1164/rccm.201707-1434OC. PubMed PMID: 29313708; PubMed Central PMCID: PMC5955059.
26. Bell JT, Tsai PC, Yang TP, Pidsley R, Nisbet J, Glass D, et al. Epigenome-wide scans identify differentially methylated regions for age and age-related phenotypes in a healthy ageing population. *PLoS Genet*. 2012;8(4):e1002629. Epub 2012/04/26. doi: 10.1371/journal.pgen.1002629. PubMed PMID: 22532803; PubMed Central PMCID: PMC3330116.
27. Husquin LT, Rotival M, Fagny M, Quach H, Zidane N, McEwen LM, et al. Exploring the genetic basis of human population differences in DNA methylation and their causal impact on immune gene regulation. *Genome Biology*. 2018;19. doi: ARTN 222. 10.1186/s13059-018-1601-3. PubMed PMID: WOS:000453758900001.
28. Hawe JS, Wilson R, Schmid KT, Zhou L, Lakshmanan LN, Lehne BC, et al. Genetic variation influencing DNA methylation provides insights into molecular mechanisms regulating genomic function. *Nature Genetics*. 2022;54(1):18-+. doi: 10.1038/s41588-021-00969-x. PubMed PMID: WOS:000737754100002.
29. Dobbyn A, Huckins LM, Boocock J, Sloofman LG, Glicksberg BS, Giambartolomei C, et al. Landscape of Conditional eQTL in Dorsolateral Prefrontal Cortex and Co-localization with Schizophrenia GWAS. *Am J Hum Genet*. 2018;102(6):1169-84. Epub 2018/05/29. doi: 10.1016/j.ajhg.2018.04.011. PubMed PMID: 29805045; PubMed Central PMCID: PMC5993513.
30. Consortium GT. The GTEx Consortium atlas of genetic regulatory effects across human tissues. *Science*. 2020;369(6509):1318-30. Epub 2020/09/12. doi: 10.1126/science.aaz1776. PubMed PMID: 32913098; PubMed Central PMCID: PMC7737656.
31. Zhang Z, Luo K, Zou Z, Qiu M, Tian J, Sieh L, et al. Genetic analyses support the contribution of mRNA N(6)-methyladenosine (m(6)A) modification to human disease heritability. *Nat Genet*. 2020;52(9):939-49. Epub 2020/07/01. doi: 10.1038/s41588-020-0644-z. PubMed PMID: 32601472; PubMed Central PMCID: PMC7483307.
32. Bell JT, Pai AA, Pickrell JK, Gaffney DJ, Pique-Regi R, Degner JF, et al. DNA methylation patterns associate with genetic and gene expression variation in HapMap cell lines. *Genome Biol*. 2011;12(1):R10. Epub 2011/01/22. doi: 10.1186/gb-2011-12-1-r10. PubMed PMID: 21251332; PubMed Central PMCID: PMC3091299.

33. Storey JD, Tibshirani R. Statistical significance for genomewide studies. *Proc Natl Acad Sci U S A*. 2003;100(16):9440-5. Epub 2003/07/29. doi: 10.1073/pnas.1530509100. PubMed PMID: 12883005; PubMed Central PMCID: PMCPMC170937.
34. Gaunt TR, Shihab HA, Hemani G, Min JL, Woodward G, Lyttleton O, et al. Systematic identification of genetic influences on methylation across the human life course. *Genome Biol*. 2016;17:61. Epub 2016/04/03. doi: 10.1186/s13059-016-0926-z. PubMed PMID: 27036880; PubMed Central PMCID: PMCPMC4818469.
35. Fan Y, Vilgalys TP, Sun S, Peng Q, Tung J, Zhou X. IMAGE: high-powered detection of genetic effects on DNA methylation using integrated methylation QTL mapping and allele-specific analysis. *Genome Biol*. 2019;20(1):220. Epub 2019/10/28. doi: 10.1186/s13059-019-1813-1. PubMed PMID: 31651351; PubMed Central PMCID: PMCPMC6813132.
36. Ng MCY, Graff M, Lu YC, Justice AE, Mudgal P, Liu CT, et al. Discovery and fine-mapping of adiposity loci using high density imputation of genome-wide association studies in individuals of African ancestry: African Ancestry Anthropometry Genetics Consortium. *Plos Genetics*. 2017;13(4). doi: ARTN e1006719
10.1371/journal.pgen.1006719. PubMed PMID: WOS:000402549200032.
37. Liang J, Le TH, Edwards DRV, Tayo BO, Gaulton KJ, Smith JA, et al. Single-trait and multi-trait genome-wide association analyses identify novel loci for blood pressure in African-ancestry populations. *PLoS Genet*. 2017;13(5):e1006728. Epub 2017/05/13. doi: 10.1371/journal.pgen.1006728. PubMed PMID: 28498854; PubMed Central PMCID: PMCPMC5446189.
38. Jamieson E, Korologou-Linden R, Wootton RE, Guyatt AL, Battram T, Burrows K, et al. Smoking, DNA Methylation, and Lung Function: a Mendelian Randomization Analysis to Investigate Causal Pathways. *Am J Hum Genet*. 2020;106(3):315-26. Epub 2020/02/23. doi: 10.1016/j.ajhg.2020.01.015. PubMed PMID: 32084330; PubMed Central PMCID: PMCPMC7058834.
39. Hatcher C, Relton CL, Gaunt TR, Richardson TG. Leveraging brain cortex-derived molecular data to elucidate epigenetic and transcriptomic drivers of complex traits and disease. *Transl Psychiatry*. 2019;9(1):105. Epub 2019/03/02. doi: 10.1038/s41398-019-0437-2. PubMed PMID: 30820025; PubMed Central PMCID: PMCPMC6395652.
40. Guan YT, Liang XJ, Ma ZY, Hu HL, Liu HB, Miao Z, et al. A single genetic locus controls both expression of DPEP1/CHMP1A and kidney disease development via ferroptosis. *Nature Communications*. 2021;12(1). doi: ARTN 5078
10.1038/s41467-021-25377-x. PubMed PMID: WOS:000687672000009.
41. Soliai MM, Kato A, Helling BA, Stanhope CT, Norton JE, Naughton KA, et al. Multi-omics colocalization with genome-wide association studies reveals a context-specific genetic mechanism at a childhood onset asthma risk locus. *Genome Med*. 2021;13(1):157. Epub 2021/10/12. doi: 10.1186/s13073-021-00967-y. PubMed PMID: 34629083; PubMed Central PMCID: PMCPMC8504130 grants from the NIH; is a paid consultant for AstraZeneca, Meissa Vaccines Inc., and Gossamer Bio; and has stock options in Meissa Vaccines Inc. D.J.J. declares personal fees from Novartis, GSK, Pfizer, Sanofi, Regeneron, Astra Zeneca, and Vifor Pharma. Grant funding from NIAID, NHLBI, and GSK. J.C.C. received research materials from Pharmavite (vitamin D and placebo capsules) and Merck and GSK (inhaled steroids) in order to provide medications free of cost to participants in NIH-funded studies, unrelated to the current work.

R.P.S. reports consulting fees from Intersect ENT, Merck, GlaxoSmithKline, Sanofi, AstraZeneca/Medimmune, Genentech, Actobio Therapeutics, Lyra Therapeutics, Astellas Pharma, Allakos, and Otsuka. R.P.S. also receives royalties from Siglec-8 and Siglec-8 ligand-related patents licensed by Johns Hopkins to Allakos Inc. The remaining authors declare that they have no competing interests.

42. Hannon E, Gorrie-Stone TJ, Smart MC, Burrage J, Hughes A, Bao Y, et al. Leveraging DNA-Methylation Quantitative-Trait Loci to Characterize the Relationship between Methyloomic Variation, Gene Expression, and Complex Traits. *Am J Hum Genet.* 2018;103(5):654-65. Epub 2018/11/08. doi: 10.1016/j.ajhg.2018.09.007. PubMed PMID: 30401456; PubMed Central PMCID: PMC6217758.

43. Ek WE, Rask-Andersen M, Karlsson T, Enroth S, Gyllensten U, Johansson A. Genetic variants influencing phenotypic variance heterogeneity. *Hum Mol Genet.* 2018;27(5):799-810. Epub 2018/01/13. doi: 10.1093/hmg/ddx441. PubMed PMID: 29325024.

44. Kulkarni H, Kos MZ, Neary J, Dyer TD, Kent JW, Jr., Goring HH, et al. Novel epigenetic determinants of type 2 diabetes in Mexican-American families. *Hum Mol Genet.* 2015;24(18):5330-44. Epub 2015/06/24. doi: 10.1093/hmg/ddv232. PubMed PMID: 26101197; PubMed Central PMCID: PMC4550817.

45. Villicana S, Bell JT. Genetic impacts on DNA methylation: research findings and future perspectives. *Genome Biol.* 2021;22(1):127. Epub 2021/05/02. doi: 10.1186/s13059-021-02347-6. PubMed PMID: 33931130; PubMed Central PMCID: PMC8086086.

46. Bundo M, Sunaga F, Ueda J, Kasai K, Kato T, Iwamoto K. A systematic evaluation of whole genome amplification of bisulfite-modified DNA. *Clin Epigenetics.* 2012;4(1):22. Epub 2012/11/24. doi: 10.1186/1868-7083-4-22. PubMed PMID: 23174095; PubMed Central PMCID: PMC3536718.

47. Vieira-Lara MA, Dommerholt MB, Zhang W, Blankestijn M, Wolters JC, Abegaz F, et al. Age-related susceptibility to insulin resistance arises from a combination of CPT1B decline and lipid overload. *BMC Biol.* 2021;19(1):154. Epub 2021/08/01. doi: 10.1186/s12915-021-01082-5. PubMed PMID: 34330275; PubMed Central PMCID: PMC8323306.

48. Parsinen J, Kuukasjarvi T, Karhu R, Kallioniemi A. High-level amplification at 17q23 leads to coordinated overexpression of multiple adjacent genes in breast cancer. *Brit J Cancer.* 2007;96(8):1258-64. doi: 10.1038/sj.bjc.6603692. PubMed PMID: WOS:000245831800016.

49. Liu M, Liu P, Chang Y, Xu B, Wang N, Qin L, et al. Genome-wide DNA methylation profiles and small noncoding RNA signatures in sperm with a high DNA fragmentation index. *J Assist Reprod Genet.* 2022;39(10):2255-74. Epub 2022/10/04. doi: 10.1007/s10815-022-02618-6. PubMed PMID: 36190595; PubMed Central PMCID: PMC9596664.

50. Gracilla DE, Korla PK, Lai MT, Chiang AJ, Liou WS, Sheu JJ. Overexpression of wild type or a Q311E mutant MB21D2 promotes a pro-oncogenic phenotype in HNSCC. *Mol Oncol.* 2020;14(12):3065-82. Epub 2020/09/27. doi: 10.1002/1878-0261.12806. PubMed PMID: 32979859; PubMed Central PMCID: PMC7718949.

51. Huan T, Joehanes R, Song C, Peng F, Guo Y, Mendelson M, et al. Genome-wide identification of DNA methylation QTLs in whole blood highlights pathways for cardiovascular disease. *Nat Commun.* 2019;10(1):4267. Epub 2019/09/21. doi: 10.1038/s41467-019-12228-z. PubMed PMID: 31537805; PubMed Central PMCID: PMC6753136.

52. Bibikova M, Le J, Barnes B, Saedinia-Melnyk S, Zhou LX, Shen R, et al. Genome-wide DNA methylation profiling using Infinium (R) assay. *Epigenomics*. 2009;1(1):177-200. doi: 10.2217/Epi.09.14. PubMed PMID: WOS:000278041000021.

REVIEWERS' COMMENTS

Reviewer #1 (Remarks to the Author):

the author has address all of the comments.

Reviewer #2 (Remarks to the Author):

Authors mentioned that due to computational burden and storage limitations, they were unable to perform the trans-meQTL analysis. I feel it is ok that in this paper, they focused on cis only. However, I hope they can continue to explore trans in this dataset in the near future. Because trans meQTLs will be a valuable resource.